# Linear Regression using Heterogeneous Data Batches

**Ayush Jain**[*]
Granica Computing Inc.
ayush.jain@granica.ai

**Rajat Sen**
Google Research
senrajat@google.com

**Weihao Kong**
Google Research
weihaokong@google.com

**Abhimanyu Das**
Google Research
abhidas@google.com

**Alon Orlitsky**
UC San Diego
alon@ucsd.edu

## Abstract

In many learning applications, data are collected from multiple sources, each providing a *batch* of samples that by itself is insufficient to learn its input-output relationship. A common approach assumes that the sources fall in one of several unknown subgroups, each with an unknown input distribution and input-output relationship. We consider one of this setup's most fundamental and important manifestations where the output is a noisy linear combination of the inputs, and there are $k$ subgroups, each with its own regression vector. Prior work [KSS$^+$20] showed that with abundant small-batches, the regression vectors can be learned with only few, $\tilde{\Omega}(k^{3/2})$, batches of medium-size with $\tilde{\Omega}(\sqrt{k})$ samples each. However, the paper requires that the input distribution for all $k$ subgroups be isotropic Gaussian, and states that removing this assumption is an "interesting and challenging problem". We propose a novel gradient-based algorithm that improves on the existing results in several ways. It extends the applicability of the algorithm by: (1) allowing the subgroups' underlying input distributions to be different, unknown, and heavy-tailed; (2) recovering all subgroups followed by a significant proportion of batches even for infinite $k$; (3) removing the separation requirement between the regression vectors; (4) reducing the number of batches and allowing smaller batch sizes.

## 1 Introduction

In numerous applications, including federated learning [WCX$^+$21], sensor networks [WZ89], crowd-sourcing [SVC16] and recommendation systems [WDVR06], data are collected from multiple sources, each providing a *batch* of samples. For instance, in movie recommendation systems, users typically rate multiple films. Since all samples in a batch are generated by the same source, they are often assumed to share the same underlying distribution. However, the batches are frequently very small, e.g., many users provide only few ratings. Hence, it may be impossible to learn a different model for each batch.

A common approach has therefore assumed [TLW99] that all batches share the same underlying distribution and learn this common model by pooling together the data from all batches. While this may work well for some applications, in others it may fail, or lack personalization. For instance, in recommendation systems, it may not capture the characteristics of individual users.

A promising alternative that allows for personalization even with many small batches, assumes that batches can be categorized into $k$ sub-populations with similar underlying distributions. Hence in

---

[*]This work was done while the author was a student at UCSD and interning part-time at Google Research.

38th Conference on Neural Information Processing Systems (NeurIPS 2024).

each sub-population, all underlying distributions are close to and can be represented by a single distribution. Even when $k$ is large, our work allows the recovery of models for sub-populations with a significant fraction of batches. For example, in the recommendation setting, most users can be classified into a few sub-populations such that the distribution of users in the sub-population is close, for instance, those preferring certain genres.

In this paper, we focus on the canonical model of linear regression in supervised learning. A distribution $\mathcal{D}$ of samples $(x, y)$ follows a linear regression model if, for some regression vector $w \in \mathbb{R}^d$, the output is $y = w \cdot x + \eta$ where the input $x$ is a random $d$ dimensional vector and $\eta$ is a zero-mean noise. While we allow the presence of sub-populations that do not follow the linear regression model or have very few batches, our goal is to recover the regression vectors for all large sub-populations that follow this linear regression model.

## 1.1 Our Results

This setting was first considered in [KSS$^+$20] for meta-learning applications, where they view and term batches as *tasks*. [KSS$^+$20] argue that in meta-learning applications task or batch lengths follow a long tail distribution and in the majority of the batches only a few labeled examples are available. Only a few batches have medium-size labeled samples available, and almost all of them have length $\ll d$. Note that similar observations have been made in the recommender system literature where the distribution of a number of ratings per user follows a long-tailed distribution with an overwhelming number of users rating only a few items while rare tail users rating hundreds of items [GKG15]. The same has been observed for the distribution of the number of ratings per item [PT08]. Therefore, it is reasonable to assume that in these applications of interest, a handful of medium-size batches along with a large number of batches of constant size are available. Under this setting our main results allow recovery of all sub-populations that has a significant fraction of batches and follow a linear regression model.

Building upon the preceding discussion, we have two sets of batches. Each batch comprises i.i.d. samples drawn from one of the $k$ sub-populations. The distribution of samples for these sub-populations is unknown, as is the identity of the sub-population from which each batch originates. Batches in set $B_s$ (denoted as *small*) consist of at least two samples each, while those in set $B_m$ (denoted as *medium*) are of larger size, containing $\geq n_m$ samples. Let's consider a sub-population indexed by $i$, which follows a linear regression model with parameter vector $w_i$ and output-noise variance $\leq \sigma^2$. The next theorem presents the guarantees provided by our algorithm for estimating $w_i$.

**Theorem 1.1** (Informal). *Let $\alpha > 0$ such that both sets $B_s$ and $B_m$ comprises at least $\alpha$ fraction of batches from a sub-population $i$. Given $|B_s| \geq \tilde{\Omega}(d/\alpha^2)$, $|B_m| \geq \tilde{\Omega}(\min(\sqrt{k}, 1/\sqrt{\alpha})/\alpha)$, and $n_m \geq \tilde{\Omega}(\min(\sqrt{k}, 1/\sqrt{\alpha}))$, our algorithm runs in time $poly(d, 1/\alpha, k)$ and outputs a list $L$ of size $\tilde{O}(1/\alpha)$ such that w.h.p., at least one estimate in $L$ is within a distance of $o(\sigma)$ from $w_i$ and has an expected prediction error $\sigma^2(1 + o(1))$ for sub-population $i$. Moreover, given $\Omega(\log L)$ samples from sub-population $i$, we can identify such an estimate from $L$.*

In the above theorem, the batches within set $B_s$ must contain a minimum of 2 samples, while those within set $B_m$ must be of size $\geq n_m$. Additionally, both sets must include $\geq \alpha$ fraction of batches originating from the population we want to recover. It follows that our algorithm requires only $\alpha(2|B_s| + n_m|B_s|) = \tilde{\Omega}(d/\alpha)$ samples from that specific sub-population. It's worth noting that any algorithm, even when dealing with a single sub-population ($k = 1$), requires a minimum of $\Omega(d)$ samples from said sub-population. To the best of our knowledge, ours is the best sample complexity for recovering the linear regression models in the presence of multiple sub-populations using batch sizes smaller than $d$.

*Remark* 1.1. When multiple sub-populations meet the criteria outlined for sub-population $i$, the list $L$ returned by our algorithm includes estimates of the regression vector for each of these sub-populations. Furthermore, since there can be up to $\min\{k, 1/\alpha\}$ distinct sub-populations that satisfy these criteria, any algorithm that aims to provide meaningful guarantees must produce a list of size $\geq \min\{k, 1/\alpha\}$, unless the algorithm has access to a sample set known to originate from the specific sub-population $i$ it aims to recover. A similar situation arises in the list-decodable setup [CSV17], where also the algorithm returns a list of estimates.

## 1.2  Comparison to Prior Work

The only work that provides a polynomial-time algorithm in dimension $d$, in the same generality as ours is [DJKS22]. They even allow the presence of adversarial batches. However, they require $\tilde{\Omega}(d/\alpha^2)$ batches from the sub-population, each of them of size $\tilde{\Omega}(1/\alpha)$, and therefore, $\tilde{\Omega}(d/\alpha^3)$ samples in total, which exceeds our sample complexity by a factor of $1/\alpha^2$. Furthermore, their algorithm requires a minimum number of samples per batch, which is quadratically larger than ours. All other works place strong assumptions on the distributions of the sub-population and still require a number of samples much larger than ours, which we discuss next.

Most of the previous works [CL13, SJA16, ZJD16, YCS16, LL18, CLS20, DK20, PMSG22] have addressed the widely studied *mixed linear regression (MLR)* model where all batches are of size 1, and adhere to the following three assumptions:

1. All $k$ sub-populations have $\geq \alpha$ fraction of data. This assumption implies $k \leq 1/\alpha$.

2. All $k$ distributions follow a linear regression model.

3. All $k$ regression coefficients are well separated, namely $\|w_i - w_j\| \geq \Delta, \forall\, i \neq j$ .

Even for $k = 2$, solving MLR, in general, is NP-hard [YCS14]. Hence all these works on mixed linear regression, except [LL18], also made the following assumption:

4. All input distributions (i.e., the distribution over $x$) are the same for every sub-population, in fact, the same isotropic Gaussian distribution.

With this additional isotropic Gaussian assumption, they provided algorithms that have runtime and sample complexity polynomial in the dimension. However, even with these four strong assumptions, their sample complexity is super-polynomial overall. In particular, the sample complexity in [ZJD16, CLS20, DK20] is quasi-polynomial in $k$ and [CL13, SJA16, DK20] require at least a quadratic scaling in $d$. In [CL13, SJA16, YCS16] the sample complexity scales as a large negative power of the minimum singular value of certain moment matrix of regression vectors that can be zero even when the gap between the regression vectors is large. In addition, [ZJD16, YCS16, CLS20] required zero-noise i.e $\eta = 0$. The only work we are aware of that can avoid Assumption 4 and handle different input distributions for different sub-populations under MLR is [LL18]. However, they still require all distributions to be Gaussian and $\eta = 0$, and their sample size, and hence run-time is exponential, $\Omega(\exp(k^2))$ in $k$. Recently, [TV23] proposed and analyzed an approximate message-passing algorithm for mixed linear regression. However, their analysis is asymptotic, providing convergence guarantees as the number of samples and dimensions tends to infinity, while assuming a finite number of components. The results also rely on Assumptions 2 and 4.

The work that most closely relates to ours is [KSS$^+$20], which considers batch sizes $> 1$. While it achieves the same dependence as us on $d, k$, and $1/\alpha$, on the length and number of medium and small batches, the sample complexity of the algorithms and the minimum length required for medium-size batches had an additional multiplicative dependence on the inverse separation parameter $1/\Delta$. It also required Assumption 4 mentioned in the section. The follow-up work [KSKO20] which still assumes all four assumptions can handle the presence of a small fraction $\ll \alpha^2/k^2$ of adversarial batches, but requires $\tilde{\Omega}(dk^2/\alpha^2 + k^5/\alpha^4)$ samples. It also suffers from strong assumptions similar to those in earlier work, and the sum-of-squares approach makes it impractical. The sum of the square approach, and stronger isotropic Gaussian assumption, allow it to achieve a better dependence on $1/\alpha$ on medium-size batch lengths, however, causing a significant increase in the number of medium-size batches required.

**Our improvement over prior work.** In contrast, our work avoids all four assumptions, and can recover any sufficiently large sub-populations that follow a linear regression model. In particular: (1) Even when a large number of different sub-populations are present, (e.g., $k \geq 1/\alpha$), we can still recover the regression coefficient of a sub-population with sufficient fraction of batches. (2) The $k$ distributions do not even need to follow a linear regression model. In particular, our algorithm is robust to the presence of other sub-populations for which the conditional distribution of output given input is arbitrary. (3) Our work requires no assumption on the separation of regression coefficient $\Delta$, and our guarantees as well have no dependence on the separation. (4) We allow different input distributions for different sub-populations, which can be unknown and heavy tailed, (5) In addition

to removing the four assumptions, the algorithm doesn't require all batches in a sub-population to have identical distributions, it only requires them to be close (see Remark 2.1).

## 1.3 Techniques and Organization

To recover the regression vector for a sub-population $i$ that follow the linear regression model and has more than $\alpha$ fraction of batches, the algorithm begins by selecting a medium size batch from this sub-population. To obtain such a batch we randomly choose $\tilde{\Theta}(1/\alpha)$ batches from the set of medium-size batches $B_m$ and repeat our algorithm to obtain one estimate for every batch chosen. With high probability, at least one of the randomly selected medium-sized batches belongs to sub-population $i$, and when the algorithm is run for such a batch, w.h.p. it recovers the estimate of $w_i$. Therefore, at least one of the $\tilde{\Theta}(1/\alpha)$ estimates accurately recover $w_i$.

The regression vector $w_i$ minimizes the expected squared loss for the sub-population $i$. Therefore, we use a gradient-descent-based approach to estimate $w_i$. We start with an initial estimate (all zero) and improve this estimate by performing multiple rounds of gradient descent steps.

First, using a large number of smaller batches we estimate a smaller subspace of $\mathbb{R}^d$ that preserves the norm of the gradient for sub-population $i$. Next, using the medium-size batch selected at the beginning, we test which of the other medium-size batches have a projection of gradient close to the chosen batch, and use them to estimate the gradient in the smaller subspace. The advantage of sub-space reduction is that testing and estimation of the gradient in the smaller subspace is easier, consequently reducing the minimum length of medium-size batches required for testing and the number of medium-size batches required for estimation. Our gradient estimation approach draws inspiration from [KSS⁺20]. However, while they employed it to directly estimate regression vectors of all sub-populations simultaneously , we adapt it to sequentially estimate gradients for one sub-population at a time. A crucial new ingredient of our algorithm is the clipping of gradients, which limits the effect of other components and allows the algorithm to work even for heavy-tailed distributions.

We describe the algorithm in detail in Section 3 after having presented our main theorems in Section 2. Then in Section 4 we compare our algorithm with the one in [KSS⁺20] on simulated datasets, to show that our algorithm performs better in the setting of the latter paper as well as generalizes to settings that are outside the assumptions of [KSS⁺20].

# 2 Problem Formulation and Main Results

Consider distributions $\mathcal{D}_0, \ldots, \mathcal{D}_{k-1}$ over input-output pairs $(x, y) \in \mathbb{R}^d \times \mathbb{R}$ each corresponding to one of the $k$ sub-populations. A *batch* $b$ consists of i.i.d. samples from one of the distributions. Samples in different batches are independent. There are two sets of batches. The batches in $B_s$ are *small* and contain at least two samples each, while batches in $B_m$ are of medium size and contain at least $n_m$ samples. For any batch the identity of the distribution it is sampled from is unknown. Next, we describe the distributions. To aid it and the remaining paper we first introduce some notation.

The $L_2$ *norm* of a vector $u$ is denoted by $\|u\|$. The *norm*, or *spectral norm*, of a matrix $M$ is denoted by $\|M\|$ and is defined as $\max_{\|u\|=1} \|Mu\|$. If $M$ is a symmetric matrix, the norm simplifies to $\|M\| = \max_{\|u\|=1} |u^\intercal M u|$. The trace of a symmetric matrix $M$ is $\mathrm{Tr}(M) := \sum_i M_{ii}$, the sum of the elements on the main diagonal of $M$. We will use the symbol $S$ to denote an arbitrary collection of samples. For a batch denoted by $b$, we will use $S^b$ to represent the set of all samples in the batch.

## 2.1 Data Distributions

Let $\Sigma_i := \mathbb{E}_{\mathcal{D}_i}[xx^\intercal]$ denote the second-moment matrix of input for distribution $\mathcal{D}_i$.

To recover the regression vector for a distribution $\mathcal{D}_i$, we assume that it satisfies the following assumptions[2]:

1. (Input distribution) There are constants $C$ and $C_1$ such that for all $i \in I$,

---

[2]We note that these assumptions are standard in heavy-tailed linear regression where some condition on the anti-concentration of $\mathbb{E}[xx^\intercal]$ is necessary to achieve finite sample complexities [LM16, Oli16, CAT⁺20], even in the specific case of a single sub-population.

(a) *L4-L2* hypercontractivity: For all $u \in \mathbb{R}^d$, $\mathbb{E}_{\mathcal{D}_i}[(x \cdot u)^4] \leq C(\mathbb{E}_{\mathcal{D}_i}[(x \cdot u)^2])^2$.

(b) Bounded condition number: For normalization purpose, we assume $\min_{\|u\|=1} u^\intercal \Sigma_i u \geq 1$ and to bound the condition number we assume that $\|\Sigma_i\| \leq C_1$.

2. (Input-output relation) There is a $\sigma > 0$ s.t. for all $i \in I$, $y = w_i \cdot x + \eta$, where $w_i \in \mathbb{R}^d$ is an unknown regression vector, and $\eta$ is a noise independent of $x$, with zero mean $\mathbb{E}_{\mathcal{D}_i}[\eta] = 0$, and $\mathbb{E}_{\mathcal{D}_i}[\eta^2] \leq \sigma^2$. Note that by definition, the distribution of $\eta$ may differ for each $i$.

For a given $\alpha_s > 0$ and $\alpha_m > 0$, our goal is to estimate the regression vector $w_i$ for all sub-populations that have at least $\alpha_s$ and $\alpha_m$ fractions of the batches within $B_s$ and $B_m$, respectively, and follow the assumptions described above. Let $I \subseteq \{0, 1, .., k-1\}$ denote the collection of indices corresponding to all such sub-populations. (Note that in the introduction, we simplified this more general scenario by assuming equal values for $\alpha_s$ and $\alpha_m$, both denoted by $\alpha$.)

For other sub-populations $i \notin I$, we only need that the input distribution satisfies $\|\Sigma_i\| \leq C_1$, same as the second half of assumption 1(b). The input-output relation for samples generated by $\mathcal{D}_i$ for $i \notin I$ *can be arbitrary*, and in particular, does not even need to follow a linear regression model. Moreover, the fraction of batches from these other sub-populations can also be arbitrary.

To simplify the presentation, we make two additional assumptions. First, there is a constant $C_2 > 0$ such that for all components $i \in \{0, 1, .., k-1\}$, and random sample $(x, y) \sim \mathcal{D}_i$, $\|x\| \leq C_2 \sqrt{d}$, a.s. Second, for all $i \in I$ and a random sample $(x, y) \sim \mathcal{D}_i$, the noise distribution $\eta = y - w_i \cdot x$ is symmetric around 0. As discussed in Appendix J, these assumptions can be easily removed.

*Remark* 2.1. For simplicity, we assumed that each batch exactly follows one of the $k$ distributions. However, our methods can be extended to more general scenarios. Let $\mathcal{D}^b$ denote the underlying distribution of batch $b$. Instead of requiring $\mathcal{D}^b = \mathcal{D}_i$ for some $i \in \{0, 1, .., k-1\}$, our approach can be extended to cases where the expected gradients for $\mathcal{D}^b$ and $\mathcal{D}_i$ are close. Additionally, if $i \in I$, the regression vector $w_i$ must achieve a mean squared error (MSE) of at most $\sigma^2$ for $\mathcal{D}^b$. These conditions are satisfied if the following hold: (1) $\|\mathbb{E}_{\mathcal{D}^b}[xx^\intercal] - \Sigma_i\|$ is small, (2) for all $x \in \mathbb{R}^d$, $|\mathbb{E}_{\mathcal{D}^b}[y|x] - \mathbb{E}_{\mathcal{D}_i}[y|x]|$ is small, and (3) if $i \in I$ then for all $x \in \mathbb{R}^d$, $\mathbb{E}_{\mathcal{D}^b}[(y - w_i \cdot x)^2|x] \leq \sigma^2$. Thus, the strict identity requirement $\mathcal{D}^b = \mathcal{D}_i$ can therefore be replaced by these three approximation conditions in our analysis. This extension is feasible as our algorithm and its analysis are based on gradient descent, and can handle stochastic noise. Since the analysis tolerates statistical noise, it can also accommodate small deviations in the gradient distributions across batches in the same sub-population.

## 2.2 Main Results

We begin by presenting our result for estimating the regression vector of a component $\mathcal{D}_i$, for any $i \in I$. This result assumes that in addition to the batch collections $B_s$ and $B_m$, we have an extra medium-sized batch denoted as $b^*$ which contains samples from $\mathcal{D}_i$ and w.l.o.g, we assume $i = 0$.

**Theorem 2.1.** *Suppose index $0$ is in set $I$ and let $b^*$ be a batch of $\geq n_m$ i.i.d. samples from $\mathcal{D}_0$. Given $\delta, \epsilon \in (0, 1]$, $|B_s| = \tilde{\Omega}(\frac{d}{\alpha_s^2 \epsilon^4})$, $n_m = \tilde{\Omega}(\min\{\sqrt{k}, \frac{1}{\epsilon\sqrt{\alpha_s}}\} \cdot \frac{1}{\epsilon^2})$, and $|B_m| = \tilde{\Omega}(\frac{1}{\alpha_m}\min\{\sqrt{k}, \frac{1}{\epsilon\sqrt{\alpha_s}}\})$, Algorithm 1 runs in polynomial time and returns estimate $\hat{w}$, such that with probability $\geq 1 - \delta$, $\|\hat{w} - w_0\| \leq \epsilon\sigma$.*

We provide a proof sketch of Theorem 2.1 and the description of Algortihm 1 in Section 3, and a formal proof in Appendix H. Algorithm 1 can be used to estimate $w_i$ for all $i \in I$, and the requirement of a separate batch $b^*$ is not crucial. As mentioned in the introduction, it can be obtained by repeatedly sampling a batch from $B_m$ and running the algorithm for these sampled $b^*$. Since all the components in $I$ have $\geq \alpha_m$ fraction of batches in $B_m$, then randomly sampling $b^*$ from $B_m$, $\tilde{\Theta}(1/\alpha_m)$ times would ensure that, with high probability, we have $b^*$ corresponding to each component. We can then return a list of size $\tilde{\Theta}(1/\alpha_m)$ containing estimates corresponding to each sampled $b^*$. Then, with high probability, the list will have an estimate of the regression vectors for all components. Note that in this case, returning a list is unavoidable as there is no way to assign an appropriate index to the regression vector estimates. The following corollary follows from the above discussion and Theorem 2.1.

**Corollary 2.2.** *For given $\delta, \epsilon \in (0, 1]$, if $|B_s|$, $n_m$, and $|B_m|$, meet the requirements outlined in Theorem 2.1, then the above modification of Algorithm 1 runs in polynomial-time and outputs a*

*list $L$ of size $\tilde{\mathcal{O}}(1/\alpha_m)$ such that with probability $\geq 1 - \delta$, the list has an accurate estimate for regression vectors $w_i$ for each $i \in I$, namely $\max_{i \in I} \min_{\hat{w} \in L} \|\hat{w} - w_i\| \leq \epsilon \sigma$.*

In particular, this corollary implies that for any $i \in I$, the algorithm requires only $\tilde{\Omega}(d/\alpha_s)$ batches of size two and $\tilde{\Omega}(\min\{\sqrt{k}, \frac{1}{\sqrt{\alpha_s}}\})$ medium-size batches of size $\tilde{\Omega}(\min\{\sqrt{k}, \frac{1}{\sqrt{\alpha_s}}\})$ from distribution $\mathcal{D}_i$ to estimate $w_i$ within an accuracy $o(\sigma)$. Furthermore, it is easy to show that any $o(\sigma)$ accurate estimate of regression parameter $w_i$ achieves an expected prediction error of $\sigma^2(1 + o(1))$ for output $y$ given input $x$ generated from this $\mathcal{D}_i$.

Note that results work even for infinite $k$ and without any separation assumptions on regression vectors. The $\min(\sqrt{k}, 1/\sqrt{\alpha_s})$ dependence is the best of both words. This dependence is reasonable for recovering components with a significant presence or if the number is few.

The total number of samples required by the algorithm from $\mathcal{D}_i$ in small size batches $B_s$ and medium size batches $B_m$ are only $\tilde{\mathcal{O}}(d/\alpha_s)$ and $\tilde{\mathcal{O}}(\min\{k, 1/\alpha_s\})$. It is well known that any estimator would require $\Omega(d)$ samples for such estimation guarantees even in the much simpler setting with just i.i.d. data. Consequently, the total number of samples required from $\mathcal{D}_i$ by the algorithm is within $\tilde{\mathcal{O}}(1/\alpha_s)$ factor from that required in a much simpler single component setting.

The next theorem shows that for any $i \in I$, given a list $L$ containing estimate of $w_i$ and $\Omega(\log(1/\alpha_s))$ samples from $\mathcal{D}_i$, we can identify an estimate of regression vector achieving a small prediction error for $\mathcal{D}_i$. The proof of the theorem and the algorithm is in Appendix B.

**Theorem 2.3.** *Let index $i$ be in set $I$ and let $L$ be a list of $d$ dimensional vectors. Suppose for a given $\beta > 0$ at least one of the vectors in $L$ is within distance $\beta$ from the regression vector of $\mathcal{D}_i$, namely $\min_{w \in L} \|w - w_i\| \leq \beta$. Given $\mathcal{O}(\max\{1, \frac{\sigma^2}{\beta^2}\} \log \frac{L}{\delta})$ samples from $\mathcal{D}_i$ Algorithm 3 identifies an estimate $w$ from the list $L$, s.t. with probability $\geq 1 - \delta$, $\|w - w_i\| = \mathcal{O}(\beta)$ and it achieves an expected estimation error $\mathbb{E}_{\mathcal{D}_i}[(\hat{w} \cdot x - y)^2] \leq \sigma^2 + \mathcal{O}(\beta^2)$.*

*Remark* 2.2. As a special case the above theorem implies that that for any $\beta \geq \Omega(\sigma)$, Algorithm 3 can identify the near best estimate of $w_i$ from list $L$ using only $\mathcal{O}(\log \frac{L}{\delta})$ samples from $\mathcal{D}_i$.

Combining the above theorem and Corollary 2.2, we get

**Theorem 2.4.** *For given $\delta, \epsilon \in (0, 1]$, if $|B_s|, n_m$, and $|B_m|$, meet the requirements outlined in Theorem 2.1, then, there exists a polynomial-time algorithm that, with probability $\geq 1 - \delta$, outputs a list $L$ of size $\tilde{\mathcal{O}}(1/\alpha_m)$ containing estimates of $w_i$'s for $i \in I$. Furthermore, given $|S| \geq \Omega(\frac{1}{\epsilon^2} \log \frac{1}{\delta \alpha_m})$ samples from $\mathcal{D}_i$, for any $i \in I$, Algorithm 3 returns $\hat{w} \in L$ that with probability $\geq 1 - \delta$ satisfies $\|w_i - \hat{w}\| \leq \mathcal{O}(\epsilon \sigma)$ and achieves an expected estimation error $\mathbb{E}_{\mathcal{D}_i}[(\hat{w} \cdot x - y)^2] \leq \sigma^2 + \mathcal{O}(\epsilon^2 \sigma^2)$*

## 3 Algorithm for recovering regression vectors

This section provides an overview and pseudo-code of Algorithm 1, along with an outline of the proof that achieves the guarantee stated in Theorem 2.1. As per the theorem, we assume that index 0 belongs to $I$, and we have a batch $b^*$ containing $n_m$ samples from the distribution $\mathcal{D}_0$. Note that $\mathcal{D}_0$ satisfies the conditions mentioned in Section 2.1 and that $B_s$ and $B_m$ each have $\geq |B_s|\alpha_s$ and $\geq |B_m|\alpha_m$ batches with i.i.d. samples from $\mathcal{D}_0$. However, the identity of these batches is unknown.

**Gradient Descent.** Note that $w_0$ minimizes the expected square loss for distribution $\mathcal{D}_0$. Our algorithm aims to estimate $w_0$ by taking a gradient descent approach. It performs a total of $R$ gradient descent steps. Let $\hat{w}^{(r)}$ denote the algorithm's estimate of $w_0$ at the beginning of step $r$. Without loss of generality, we assume that the algorithm starts with an initial estimate of $\hat{w}^{(1)} = 0$. In step $r$, the algorithm produces an estimate $\Delta^{(r)}$ of the gradient of the expected square loss for distribution $\mathcal{D}_0$ at its current estimate $\hat{w}^{(r)}$. For convenience, we refer to this expected gradient for $\mathcal{D}_0$ at $\hat{w}^{(r)}$ simply as *expected gradient*. The algorithm then updates its current estimate for the next round as $\hat{w}^{(r+1)} = \hat{w}^{(r)} - \Delta^{(r)}/C_1$.

The main challenge the algorithm faces is the accurate estimation of the expected gradients in each step. Accurately estimating the expected gradients at each step would require $\Omega(d/\epsilon^2)$ i.i.d. samples from $\mathcal{D}_0$. However, our algorithm only has access to a medium-size batch $b^*$ that is guaranteed to have samples from $\mathcal{D}_0$ and this batch contains far fewer samples. And for batches in $B_s$ and $B_m$,

**Algorithm 1** MainAlgorithm

---

1: **Input:** Collections of batches $B_s$ and $B_m$, $\alpha_s$, $k$, a medium size batch $b^*$ of i.i.d. samples from $\mathcal{D}_0$, distribution parameters (upper bounds) $\sigma$, $C$, $C_1$, an upper bound $M$ on $\|w_0\|$, $\epsilon$ and $\delta$,

2: **Output:** Estimate of $w_0$

3: $\epsilon_1 \leftarrow \Theta(1)$, $\epsilon_2 \leftarrow \Theta\left(\frac{1}{C_1\sqrt{C+1}}(\epsilon_1 + \frac{1}{\sqrt{C_1}})\right)$, $\ell \leftarrow \min\{k, \frac{1}{2\alpha_s \epsilon_2^2}\}$, $\delta' \leftarrow \frac{\delta}{5R}$ and $R \leftarrow \Theta(C_1 \log \frac{M}{\sigma})$

4: Partition the collection of batches $B_s$ into $R$ disjoint same size random parts $\{B_s^{(r)}\}_{r\in[R]}$.

5: Similarly partition $B_m$ into $R$ disjoint same size random parts $\{B_m^{(r)}\}_{r\in[R]}$.

6: Divide samples $S^{b^*}$ into $2R$ disjoint same size random parts $\{S_1^{b^*,(r)}\}_{r\in[R]}$ and $\{S_2^{b^*,(r)}\}_{r\in[R]}$.

7: Initialize $\hat{w}^{(1)} \leftarrow 0$

8: **for** $r$ from 1 to $R$ **do**

9: $\quad \kappa^{(r)} \leftarrow \text{CLIPEST}(S_1^{b^*,(r)}, \hat{w}^{(r)}, \epsilon_1, \delta', \sigma, C, C_1)$

10: $\quad P^{(r)} \leftarrow \text{GRADSUBEST}(B_s^{(r)}, \kappa^{(r)}, \hat{w}^{(r)}, \ell)$

11: $\quad \Delta^{(r)} \leftarrow \text{GRADEST}(B_m^{(r)}, S_2^{b^*,(r)}, \kappa^{(r)}, \hat{w}^{(r)}, P^{(r)}, \epsilon_2, \delta')$

12: $\quad \hat{w}^{(r+1)} \leftarrow \hat{w}^{(r)} - \frac{1}{C_1}\Delta^{(r)}$

13: **end for**

14: $\hat{w} \leftarrow \hat{w}^{(R+1)}$ and Return $\hat{w}$

---

the algorithm doesn't know which of the batches has samples from $\mathcal{D}_0$. Despite these challenges, we demonstrate an efficient method to estimate the expected gradients accurately.

The algorithm randomly divides sets $B_s$ and $B_m$ into $R$ disjoint equal subsets, denoted as $\{B_s^{(r)}\}_{r=1}^R$ and $\{B_m^{(r)}\}_{r=1}^R$, respectively. The samples in batch $b^*$ are divided into two collections of equal disjoint subsets, denoted as $\{S_1^{b^*,(r)}\}_{r=1}^R$ and $\{S_2^{b^*,(r)}\}_{r=1}^R$. At each iteration $r$, the algorithm uses the collections of medium and small batches $B_s^{(r)}$ and $B_m^{(r)}$, respectively, along with the two collections of i.i.d. samples $S_1^{b^*,(r)}$ and $S_2^{b^*,(r)}$ from $\mathcal{D}_0$ to estimate the gradient at point $w^{(r)}$. While this division may not be necessary for practical implementation, as shown by our simulation results later, this ensures independence between the stationary point $\hat{w}^{(r)}$ and the gradient estimate, which facilitates our theoretical analysis and only incurs a logarithmic factor in sample complexity.

Next, we describe how the algorithm estimates the gradient and the guarantees of this estimation. Due to space limitations, we provide a brief summary here, and a more detailed description, along with formal proofs, can be found in the appendix. We start by introducing a clipping operation on the gradients, which plays a crucial role in the estimation process.

**Clipping.** Recall that the squared loss of samples $(x, y)$ on point $w$ is $(x \cdot w - y)^2/2$ and its gradient is $(x \cdot w - y)x$. Instead of directly working with the gradient of the squared loss, we work with its clipped version. Given a *clipping parameter* $\kappa > 0$, the *clipped gradient* for a sample $(x, y)$ evaluated at point $w$ is defined as $\nabla f(x, y, w, \kappa) := \frac{(x \cdot w - y)}{|x \cdot w - y| \vee \kappa} \kappa x$. For a collection of samples $S$, the *clipped gradient* $\nabla f(S, w, \kappa)$ is the average of the clipped gradients of all samples in $S$, i.e., $\nabla f(S, w, \kappa) := \frac{1}{|S|} \sum_{(x,y)\in S} \nabla f(x, y, w, \kappa)$.

The clipping parameter $\kappa$ controls the level of clipping and for $\kappa = \infty$, the clipped and unclipped gradients are the same. The clipping step is necessary to make our gradient estimate more robust, by limiting the influence of the components other than $\mathcal{D}_0$ (in lemma C.1), and as a bonus, we also obtain better tail bounds for the clipped gradients. Theorem C.2 shows that for $\kappa \geq \Omega(\sqrt{\mathbb{E}_{\mathcal{D}_0}[(y - x \cdot w)^2]})$, the difference between the expected clipped gradient and the expected gradient $\|\mathbb{E}_{\mathcal{D}_0}[(\nabla f(x, y, w, \kappa)] - \mathbb{E}_{\mathcal{D}_0}[(x \cdot w - y)x]\|$ is small. Therefore, the ideal value of $\kappa$ at point $w$ is $\Theta(\sqrt{\mathbb{E}_{\mathcal{D}_0}[(y - x \cdot w)^2]})$.

For the estimate $\hat{w}^{(r)}$ at step $r$, the choice of the clipping parameter is represented by $\kappa^{(r)}$. To estimate a value for $\kappa^{(r)}$ that is close to its ideal value, the algorithm employs the subroutine CLIPEST (presented as Algorithm 4 in the appendix). The subroutine estimates the expected value of $(y - x \cdot \hat{w}^{(r)})^2$ by using i.i.d. samples $S_1^{b^*,(r)}$ from the distribution $\mathcal{D}_0$. According to Theorem D.1 in Appendix D, the subroutine w.h.p. obtains $\kappa^{(r)}$ which is close to the ideal value. This ensures that the difference between the expectation of clipped and unclipped gradients is small, and thus, estimating the expectation of clipped gradients can replace estimating the actual gradients.

**Subspace Estimation.** The algorithm proceeds by using subroutine GRADSUBEST (presented as Algorithm 5 in Appendix E) with $\widehat{B} = B_s^{(r)}$, $w = \hat{w}^{(r)}$, and $\kappa = \kappa^{(r)}$ to estimate a smaller subspace $P^{(r)}$ within $\mathbb{R}^d$. The objective of this subroutine is to ensure that, with high probability, the expected clipped gradient for distribution $\mathcal{D}_0$ at point $\hat{w}^{(r)}$ remains largely unchanged when projected onto the identified subspace. Thus, estimating the expected clipped gradient reduces to estimating its projection onto $P^{(r)}$, which requires fewer samples due to the lower dimensionality of the subspace. To achieve this, the subroutine constructs a matrix $A$ defined as $\mathbb{E}[A] = \sum_i p_i \, \mathbb{E}_{\mathcal{D}_i}[\nabla f(x,y,w,\kappa)] \, \mathbb{E}_{\mathcal{D}_i}[\nabla f(x,y,w,\kappa)]^\intercal$, where $p_i$ denotes the fraction of batches in $\widehat{B}$ that have samples from $\mathcal{D}_i$. Since $B_s^{(r)}$ are obtained by randomly partitioning $B_s$, w.h.p. $p_0 \approx \alpha_s$. It is crucial for the success of the subroutine that the expected contribution of every batch in the above expression is a PSD matrix. The clipping helps in bounding the contribution of other components and statistical noise.

The subroutine returns the projection matrix $P^{(r)}$ for the subspace spanned by the top $\ell$ singular vectors of $A$, where we choose $\ell = \min\{k, \Theta(1/\alpha_s)\}$. It is worth noting that when $1/\alpha_s$ is much smaller than $k$ (thinking of the extreme case $k = \infty$), our algorithm still only requires estimating the top $\ell = 1/\alpha_s$ dimensional subspace, since those infinitely many components can create at most $(1/\alpha_s - 1)$ directions with weight greater than $\alpha_s$. This along with clipping ensures that the direction of $\mathcal{D}_0$ must appear in the top $\Theta(1/\alpha_s)$ subspace (see Lemma E.3). Theorem E.1 in Appendix E characterizes the guarantees for this subroutine. Informally, if $\widehat{B} \geq \tilde{\Omega}(d/\alpha^2)$, then w.h.p., the expected value of the projection of the clipped gradient on this subspace is nearly the same as the expected value of the clipped gradient, namely $\| \mathbb{E}_{\mathcal{D}_0}[P^{(r)} \nabla f(x,y,w,\kappa)] - \mathbb{E}_{\mathcal{D}_0}[\nabla f(x,y,w,\kappa)] \|$ is small. We note that our construction of matrix $A$ for the subroutine is inspired by a similar construction

---

**Algorithm 2** GRADEST

1: **Input:** A collection of medium batches $\widehat{B}$, a collection of samples $S^*$ from $\mathcal{D}_0$, $\kappa$, $w$, projection matrix $P$ for subspace of $\mathbb{R}^d$, parameter $\epsilon$, $\delta'$
2: **Output:** An estimate of clipped gradient at point $w$.
3: $T_1 \leftarrow \Theta(\log \frac{|\widehat{B}|}{\delta'})$ and $T_2 \leftarrow \Theta(\log \frac{1}{\delta'})$
4: For each $b$ divide $S^b$ into two equal random parts $S_1^b$ and $S_2^b$
5: For each $b$ further divide $S_1^b$ into $2T_1$ equal random parts, and denote them as $\{S_{1,j}^b\}_{j \in [2T_1]}$
6: Divide $S^*$ into $2T_1$ equal random parts, and denote them as $\{S_j^*\}_{j \in [2T_1]}$
7: $\zeta_j^b := \left(\nabla f(S_{1,j}^b, w, \kappa) - \nabla f(S_j^*, w, \kappa)\right)^\intercal P^\intercal P \left(\nabla f(S_{1,T_1+j}^b, w, \kappa) - \nabla f(S_{T_1+j}^*, w, \kappa)\right)$
8: Let $\widetilde{B} \leftarrow \left\{ b \in \widehat{B} : median\{\zeta_j^b : j \in [T_1]\} \leq \epsilon^2 \kappa^2 C_1 \right\}$
9: For each $b$ divide $S_2^b$ into $T_2$ equal parts randomly, and denote them as $\{S_{2,j}^b\}_{j \in [T_2]}$
10: For $i \in [T_2]$, let $\Delta_i \leftarrow \frac{1}{|\widetilde{B}|} \sum_{b \in \widetilde{B}} P\nabla f(S_{2,i}^b, w, \kappa)$.
11: Let $\xi_i \leftarrow median\{j \in [T_2] : \|\Delta_i - \Delta_j\|\}$
12: Let $i^* \leftarrow \arg\min\{i \in [T_2] : \xi_i\}$ and $\Delta \leftarrow \Delta_{i^*}$
13: Return $\Delta$

---

in [KSS$^+$20], where they used it for directly estimating regression vectors. Our results generalize the applicability of the procedure to provide meaningful guarantees even when the number of components $k = \infty$. Additionally, Lemma E.3 improves matrix perturbation bounds in Lemma 5.1 of [KSS$^+$20], which is crucial for applying this procedure for heavy-tailed distributions and reducing the number of required batches.

**Estimating the expectation of clipped gradient projection.** The final subroutine, GRADEST, estimates the expected projection of the clipped gradient. It does this by utilizing two key sets: $B_m^{(r)}$, one of the $R$ partitions of medium-sized batches, and $S_2^{b^*,(r)}$, one of the $2R$ partitions of the batch $b^*$, which contains i.i.d. samples from the reference distribution $\mathcal{D}_0$. GRADEST begins by splitting each batch in $B_m^{(r)}$ into two equal parts. For each batch $b \in B_m^{(r)}$, the first part of the split is used along with samples from $\mathcal{D}_0$ in $S_2^{b^*,(r)}$ to test whether the expected projection of the clipped gradient for the distribution that generated batch $b$ is close to that of the reference distribution $\mathcal{D}_0$. With high probability, the algorithm retains batches sampled from $\mathcal{D}_0$ and rejects those from other distributions where the difference in expected projections is significant. This test requires $\tilde{\Omega}(\sqrt{\ell})$

samples in each batch, where $\ell$ is the dimension of the projected clipped gradient, which is same as the dimension of the estimated subspace in the subspace estimation space.

After identifying the relevant batches, GRADEST estimates the projection of the clipped gradients using the second half of the samples in these batches. Since the projections of the clipped gradients lie in an $\ell$ dimensional subspace, $\Omega(\ell)$ samples suffice for the estimation. To obtain high-probability guarantees, the procedure uses the median of means approach for both testing and estimation.

The guarantees of the subroutine are described in Theorem F.1, which implies that the estimate $\Delta^{(r)}$ of the gradient satisfies $\|\Delta^{(r)} - \mathbb{E}_{\mathcal{D}_0}[P^{(r)}\nabla f(x, y, w, \kappa)]\|$ is small.

**Estimation guarantees for expected gradient.** Using the triangle inequality, we have:

$$\|\Delta^{(r)} - \mathbb{E}_{\mathcal{D}_0}[(x \cdot w - y)x]\| \leq \|\mathbb{E}_{\mathcal{D}_0}[\nabla f(x, y, w, \kappa)] - \mathbb{E}_{\mathcal{D}_0}[(x \cdot w - y)x]\|$$
$$+ \|\mathbb{E}_{\mathcal{D}_0}[\nabla f(x, y, w, \kappa)] - \mathbb{E}_{\mathcal{D}_0}[P^{(r)}\nabla f(x, y, w, \kappa)]\| + \|\Delta^{(r)} - \mathbb{E}_{\mathcal{D}_0}[P^{(r)}\nabla f(x, y, w, \kappa)]\|.$$

The first term on the right represents the difference between the expectations of the gradient and the clipped gradient. The second term captures the difference between the expectation of the clipped gradient and its projection onto the estimated subspace. Finally, the last term reflects the difference between the expectation of the projection of the clipped gradient and its estimate.

All expectations are taken with respect to the distribution $\mathcal{D}_0$, and both gradients and clipped gradients are evaluated at the point $w$. Recall that at the beginning of each gradient descent step, $w$ is updated to the previous round's estimate of $w_0$.

As previously argued, all three terms on the right side of the inequality are small, which implies that $\Delta^{(r)}$ provides an accurate estimate of the expected value of the gradient for distribution $\mathcal{D}_0$. Moreover, Lemma G.1 shows that with an accurate estimation of expected gradients, gradient descent reaches an accurate estimation of $w_0$ after $\mathcal{O}(\log \frac{\|w_0\|}{\sigma})$ steps. Therefore, setting $R = \Omega(\log \frac{\|w_0\|}{\sigma})$ suffices. This completes the description of the algorithm and the proof sketch of Theorem 2.1, with a more formal proof available in Appendix H.

## 4 Empirical Results

**Setup.** We have sets $B_s$ and $B_m$ of small and medium size batches and $k$ distributions $\mathcal{D}_i$ for $i \in \{0, 1, \ldots, k-1\}$. For a subset of indices $I \subseteq \{0, 1, \ldots, k-1\}$, both $B_s$ and $B_m$ have a fraction of $\alpha$ batches that contain i.i.d. samples from $\mathcal{D}_i$ for each $i \in I$. And for each $i \in \{0, 1, \ldots, k-1\} \setminus I$ in the remaining set of indices, $B_s$ and $B_m$ have $(1 - |I|\alpha)/(k - |I|)$ fraction of batches, that have i.i.d samples from $\mathcal{D}_i$. In all figures, the output noise is distributed as $\mathcal{N}(0, 1)$. All small batches have 2 samples each, while medium-size batches have $n_m$ samples each, which we vary from 4 to 32, as shown in the plots. We fix data dimension $d = 100$, $\alpha = 1/16$, the number of small batches to $|B_s| = \min\{8dk^2, 8d/\alpha^2\}$ and the number of medium batches to $|B_m| = 256$. In all the plots, we averaged over 10 runs and report the standard error.

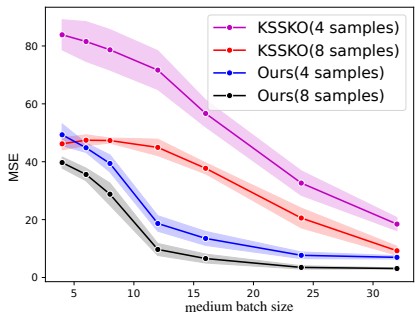
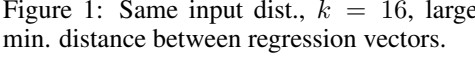
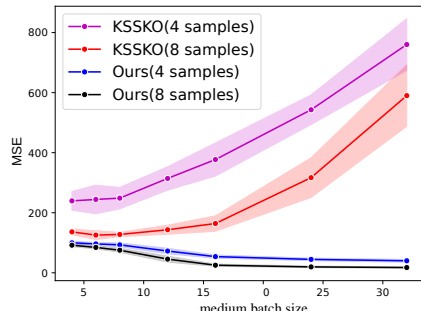

Figure 1: Same input dist., $k = 16$, large min. distance between regression vectors.

Figure 2: Different input dist., $k = 16$, large min. distance between regression vectors.

**Evaluation.** Our objective is to recover a small list containing good estimates for the regression vectors of $\mathcal{D}_i$ for each $i \in I$. We compare our proposed algorithm's performance with that of the

algorithm in [KSS+20]. Given a new batch, we can choose the weight vector from the returned list, $L$ that achieves the best error[3]. Then the MSE of the chosen weight is reported on another new batch drawn from the same distribution. The size of the new batch can be either 4 or 8 as marked in the plot. More details about our setup can be found in Appendix L.

**Setting in [KSS+20].** We first compare our algorithm with the one in [KSS+20] in the same setting as the latter paper i.e. with more restrictive assumptions. The results are displayed in Figure 1, where $I = \{0, 1, \ldots, 15\}$ and all 16 distributions have been used to generate $1/16$ fraction of the batches. All the $\mathcal{D}_i$'s are equal to $\mathcal{N}(0, I)$, and the minimum distance between the regression vectors is comparable to their norm. It can be seen that even in the original setting of [KSS+20] our algorithm significantly outperforms the other at all the different medium batch sizes plotted on the x-axis.

**Input distributions.** Our algorithm can handle different input distributions for different subgroups. We test this in our next experiment presented in Figure 2. Specifically, for each $i$, we randomly generate a covariance matrix $\Sigma_i$ such that its eigenvalues are uniformly distributed in $[1, C_1]$, and the input distribution for $\mathcal{D}_i$ is chosen as $\mathcal{N}(0, \Sigma_i)$. We set $C_1 = 4$. It can be seen that [KSS+20] completely fails in this case, while our algorithm retains its good performance.

In the interest of space, we provide additional results in Appendix L which include even more general settings: (i) when the minimum distance between regression vectors can be much smaller than their norm (ii) when the number of subgroups $k$ can be very large but the task is to recover the regression weights for the subgroups that appear in a sufficient fraction of the batches. In both these cases, our algorithm performs much better than the baseline.

## 5   Conclusion

We study the problem of learning linear regression from batched data in the presence of subpopulations. We removed several restrictive assumptions from prior work and provide better guarantees in terms of overall sample complexity. Moreover, we require relatively fewer medium batches that need to contain less number of samples compared to prior work. Finally, in our empirical results, we show that our algorithm is both practical and more performant compared to a prior baseline.

One limitation of the algorithm is that it requires knowledge of parameters $C$, $C_1$, $M$ and $\sigma$. However, note that the algorithm works even if their assumed values exceed the actual values. The change in the recovery guarantees is proportional to the disparity between the estimates and the actual parameters. The necessity of knowing reasonable upper bounds on these parameters, even for simpler problems, has been highlighted by previous work, e.g. [CAT+20].

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

# A   Other related work

**Meta Learning**. The setting we considered in this paper is closely related to *meta learning* if we treat each batch as a task. Meta-learning approaches aim to jointly learn from past experience to quickly adapt to new tasks with little available data [Sch87, TP12]. This is particularly significant in our setting when each task is associated with only a few training examples. By leveraging structural similarities among those tasks (e.g. sub-population structure), meta-learning algorithms achieve far better accuracy than what can be achieved for each task in isolation [FAL17, RL17, KZS15, OLL18, TZD+19, RRS+18]. Learning mixture of linear dynamical systems has been studied in [CP22].

**Robust and List decodable Linear Regression.** Several recent works have focused on obtaining efficient algorithms for robust linear regression and sparse linear regression when a small fraction of data may be adversarial [BJK15, BJKK17, BDLS17, Gao20, PSBR18, KKM18, DKK+19, LSLC18, KP19, DT19, MGJK19, DKS19, KKK19, PJL20, CAT+20, JLST21, KSAD22].

In scenarios where over half of the data may be arbitrary or adversarial, it becomes impossible to return a single estimate for the underlying model. Consequently, the requirement is relaxed to return a small list of estimates such that at least one of them is a good estimate for the underlying model. This relaxed framework, called "List decodable learning," was first introduced in [CSV17]. List-decodable linear regression has been studied by [KKK19, RY20, DKP+21], who have provided exponential runtime algorithms. Additionally, [DKP+21] has established statistical query lower bounds, indicating that polynomial-time algorithms may be impossible for this setting. However, as mentioned earlier, the problem can be solved in polynomial time in the batch setting as long as the batch size is greater than the inverse of the fraction of genuine data, as demonstrated in [DJKS22]. It's worth noting that an algorithm for list-decodable linear regression can be used to obtain a list of regression vector estimates for mixed linear regression.

**Robust Learning from Batches.** [QV18] presented the problem of robust learning of discrete distributions from untrustworthy batches, where a majority of batches share the same distribution and a small fraction are adversarial. They developed an exponential time algorithm for the problem. Subsequent works [CLS20] improved the run-time to quasi-polynomial, while and [JO20b] derived a polynomial time algorithm with an optimal sample complexity. The results were extended to learning one-dimensional structured distributions in [JO21, CLM20], and classification in [JO20a, KFAL20]. [AJK+22] examined a closely related problem of learning the parameters of an Erdős-Rényi random graph when a portion of nodes may be corrupted and their edges are maybe be chosen by an adversary.

# B   Selecting a regression vector from a given list

In this section, we introduce Algorithm 3 and prove that it achieves the guarantees presented in Theorem 2.3.

---

**Algorithm 3** SELECTING THE REGRESSION VECTOR

1: **Input:** Samples $S$ from $\mathcal{D}_i$ for some $i \in I$, $C_1$, a list $L$ of possible estimates of $w_i$, and $\beta \geq 0$ s.t. $\beta \geq \min_{w \in L} \|w - w_i\|$.
2: **Output:** An estimate of $w_{i*}$ from list $L$
3: Divide $S$ into $T_3 = \Theta(\log(|L|/\delta))$ equal parts $\{S_j\}_{j=1}^{T_3}$
4: **while** $\max\{\|w - w'\| : w, w' \in L\} \geq 12 C_1 \beta$ **do**
5:     pick any $w, w' \in L$ s.t. $\|w - w'\| \geq 12 C_1 \beta$.
6:     For $j \in [T_3]$, let $a_j \leftarrow \sum_{(x,y) \in S_j} \frac{1}{|S_j|} (x \cdot w - y) x \cdot (w - w')$
7:     $a \leftarrow \text{Median}\{a_j : j \in [T_3]\}$
8:     If $a > \|w - w'\|^2/4$ remove $w$, else remove $w'$ from $L$
9: **end while**
10: Return any of the remaining $w \in L$

---

Without loss of generality, assume $i = 0$, and let $w^* = \arg\min_{w \in L} \|w - w_0\|$. From the condition in the theorem, we know that $\|w^* - w_0\| \leq \beta$. The algorithm is given access to $|S| = \Omega(\max 1, \frac{\sigma^2}{\beta^2} \log \frac{|L|}{\delta})$ samples. The algorithm chooses any two vectors $w, w' \in L$ that are

more than $12C_1\beta$ distance apart and tests which of them is more likely to be within $\beta$ distance from $w_0$ using samples in $S$. The algorithm performs $T_3 = \Theta(\log \frac{|L|}{\delta})$ such tests and takes the majority vote. It retains the vector that is more likely to be closer to $w$ and discards the other from $L$. The algorithm terminates when all the vectors in $L$ are within a distance of $12C_1\beta$ from each other, by choosing a vector from those remaining in $L$ and returning it as an estimate of $w_0$. If $w^*$ is retained in $L$ until the end, using the simple triangle inequality for all $w$ that remain in $L$ at the end, we have $\|w - w_0\| \leq \|w - w^*\| + \|w^* - w_0\| \leq 12C_1\beta + \beta \leq 13C_1\beta = \mathcal{O}(\beta)$. Therefore, the estimate returned by the algorithm achieves the desired accuracy in estimating $w_0$. Hence, it suffices to show that $w^*$ is retained at the end with high probability.

Suppose $w^*$ is not in the final list $L$. Then it must have been discarded by the test in favor of $\tilde{w} \in L$ such that $\|w^* - \tilde{w}\| \geq 12C_1\beta$. The following theorem shows that for any $\tilde{w}$ such that $\|w^* - \tilde{w}\| \geq 12C_1\beta$, the probability of the testing procedure rejecting $w^*$ in favor of $\tilde{w}$ is at most $\delta/|L|$.

**Theorem B.1.** *Given $\beta > 0$, list $L$, and samples $S$ from $\mathcal{D}_0$, if $\min_{w \in L} \|w - w_0\| \leq \beta$ and $|S| = \max\{1, \frac{\sigma^2}{\beta^2}\} \log \frac{L}{\delta}$, then for the parameter $a$ computed in the while loop of Algorithm 3, with probability $1 - \delta/|L|$, we have $a \leq \|w - w'\|^2/4$ if $w = w_0$ and $a > \|w - w'\|^2/4$ if $w' = w_0$.*

The testing procedure utilized in the algorithm is based on gradients. Specifically, it calculates the average of the gradient computed on samples at point $w$ projected onto the vector $(w - w')$. The expected value of the gradient at $w$, and its projection onto $(w - w')$, are $(w - w_0)^\intercal \Sigma_0$ and $(w - w_0)^\intercal \Sigma_0 (w - w')$, respectively. If $w \approx w_0$, then the expected projection will be small. On the other hand. if $w' \approx w_0$ and $w$, then expected value of projection is $\approx (w - w')^\intercal \Sigma_0 (w - w') \gtrsim \|w - w'\|^2$. Using these observations, we prove Theorem B.1 in the next subsection.

Finally, since the maximum number of comparisons made by the algorithm is $|L| - 1$, a union bound ensures that $w^*$ will be retained until the end with probability greater than $1 - \delta$, completing the proof of Theorem 2.3.

## B.1 Proof of Theorem B.1

*Proof.* Note that $a$ is the median of the set $\{a_j : j \in [T_3]\}$, where each $a_j$ is computed using different sets of i.i.d. samples. Consequently, $\{a_j\}_{j \in [T_3]}$ are also i.i.d random variables.

We begin by calculating the expected value of $a_j$. Using the linearity of expectations, we have:

$$
\begin{aligned}
\mathbb{E}[a_j] &\overset{(a)}{=} \mathbb{E}\Big[\sum_{(x,y) \in S_j} \tfrac{1}{|S_j|}(x \cdot w - y)x \cdot (w - w')\Big] \\
&\overset{(b)}{=} \mathbb{E}_{\mathcal{D}_0}[(x \cdot w - y)x \cdot (w - w')] \\
&= \mathbb{E}_{\mathcal{D}_0}[(x \cdot (w - w_0) - (y - x \cdot w_0))x \cdot (w - w')] \\
&= \mathbb{E}_{\mathcal{D}_0}[(x \cdot (w - w_0)x \cdot (w - w')] - \mathbb{E}_{\mathcal{D}_0}[(y - x \cdot w_0)x \cdot (w - w')] \\
&= \mathbb{E}_{\mathcal{D}_0}[x \cdot (w - w_0)x \cdot (w - w')] \\
&\overset{(c)}{=} \mathbb{E}_{\mathcal{D}_0}[(x \cdot (w - w'))^2] + \mathbb{E}_{\mathcal{D}_0}[x \cdot (w' - w_0)x \cdot (w - w')],
\end{aligned}
$$

(1)
(2)

here, (a) follows from the definition of $a_j$, (b) follows from the linearity of expectation, and since $S_j$ contains i.i.d. samples from $\mathcal{D}_0$, (c) follows as the noise $y - x \cdot w_0$ has a zero mean and is independent of $x$.

Next, we compute the variance of $a_j$. Since $a_j$ represents the average of $(x \cdot w - y)x \cdot (w - w')$ over $|S_j|$ i.i.d. samples, we have

$$
\begin{aligned}
\mathrm{Var}(a_j) &= \frac{1}{|S_j|} \mathrm{Var}_{\mathcal{D}_0}((x \cdot w - y)x \cdot (w - w')) \\
&\leq \frac{1}{|S_j|} \mathbb{E}_{\mathcal{D}_0}\Big[((x \cdot w - y)x \cdot (w - w'))^2\Big].
\end{aligned}
$$

(3)

By applying Chebyshev's inequality, with a probability $\geq 3/4$, the following holds for each $a_j$:

$$
\mathbb{E}[a_j] - 2\mathrm{Var}(a_j) \leq a_j \leq \mathbb{E}[a_j] + 2\mathrm{Var}(a_j).
$$

(4)

First, we consider the case when $w = w^*$. In this case, we have $\|w_0 - w\| \leq \beta$.

Using Equation (3), we can express the variance of $a_j$ as follows:

$$\text{Var}(a_j) \leq \frac{1}{|S_j|} \mathbb{E}_{\mathcal{D}_0}\left[\left((x \cdot w - y)x \cdot (w - w')\right)^2\right]$$

$$= \frac{1}{|S_j|} \mathbb{E}_{\mathcal{D}_0}\left[\left(x \cdot (w - w_0)x \cdot (w - w') + (w_0 \cdot x - y)x \cdot (w - w')\right)^2\right]$$

$$\leq \frac{2}{|S_j|}\left(\mathbb{E}_{\mathcal{D}_0}\left[\left(x \cdot (w - w_0)x \cdot (w - w')\right)^2\right] + \mathbb{E}_{\mathcal{D}_0}\left[\left((w_0 \cdot x - y)x \cdot (w - w')\right)^2\right]\right),$$

where the last step uses the fact that for any $u, v \in \mathbb{R}$, $(u + v)^2 \leq 2u^2 + 2v^2$.

Next, we bound the two terms on the right. For the first term, we have

$$\mathbb{E}_{\mathcal{D}_0}\left[\left(x \cdot (w - w_0)x \cdot (w - w')\right)^2\right] \leq \sqrt{\mathbb{E}_{\mathcal{D}_0}[(x \cdot (w - w_0))^4]\,\mathbb{E}_{\mathcal{D}_0}[(x \cdot (w - w'))^4]}$$

$$\leq C\mathbb{E}_{\mathcal{D}_0}[(x \cdot (w - w_0))^2]\,\mathbb{E}_{\mathcal{D}_0}[(x \cdot (w - w'))^2]. \quad (5)$$

For the second term, we have:

$$\mathbb{E}_{\mathcal{D}_0}\left[\left((w_0 \cdot x - y)x \cdot (w - w')\right)^2\right] = \mathbb{E}_{\mathcal{D}_0}\left[(w_0 \cdot x - y)^2\right]\mathbb{E}_{\mathcal{D}_0}\left[\left(x \cdot (w - w')\right)^2\right]$$

$$= \sigma^2\,\mathbb{E}_{\mathcal{D}_0}\left[\left(x \cdot (w - w')\right)^2\right], \quad (6)$$

where the first inequality follows from assumption 1a and the second inequality follows from assumption 1b.

Combining the above three equations, we obtain:

$$\text{Var}(a_j) \leq \frac{2}{|S_j|} \mathbb{E}_{\mathcal{D}_0}\left[\left(x \cdot (w - w')\right)^2\right]\left(C\,\mathbb{E}_{\mathcal{D}_0}[(x \cdot (w - w_0))^2] + \sigma^2\right)$$

$$\leq \frac{2}{|S_j|}C_1\|w - w'\|^2\left(CC_1\|w - w_0\|^2 + \sigma^2\right),$$

where the last inequality uses assumption 1b.

Using Equation (1), the Cauchy-Schwarz inequality, and assumption 1b, we have:

$$\mathbb{E}[a_j] \leq C_1\|w - w_0\| \cdot \|w - w'\|. \quad (7)$$

Combining the two equations above, we obtain:

$$\mathbb{E}[a_j] + 2\sqrt{\text{Var}(a_j)} \leq \|w - w'\|\left(C_1\|w - w_0\| + \frac{2\sqrt{2C_1}}{\sqrt{|S_j|}}\left(\sigma + \sqrt{CC_1}\|w - w_0\|\right)\right)$$

$$\overset{(a)}{\leq} \|w - w'\|\left(C_1\beta + \frac{\sqrt{8C_1}}{\sqrt{|S_j|}}\sigma + \frac{\sqrt{8CC_1}}{\sqrt{|S_j|}}\beta\right)$$

$$\overset{(b)}{\leq} 3C_1\|w - w'\|\beta$$

$$\overset{(c)}{\leq} \frac{\|w - w'\|^2}{4},$$

here, in (a), we use $w = w^*$, which implies $\|w - w_0\| \leq \beta$, in (b), we utilize $|S_j| \geq 48C$ and $|S_j| \geq \frac{12\sigma^2}{C_1\beta^2}$, in (c), we use the fact that for any $w$ and $w'$ in the while loop of the algorithm, $\|w - w'\| \geq 12C_1\beta$. Consequently, it follows from Equation (4) that each $a_j \leq \frac{\|w-w'\|^2}{4}$ with probability $\geq 3/4$. Hence, with probability $\geq 1 - \delta$ the median of $a_j$ is $\leq \frac{\|w-w'\|^2}{4}$.

Next, we consider the case when $w' = w^*$. Firstly, we bound the variance using Equation (3):

$$\text{Var}(a_j) \leq \frac{1}{|S_j|} \mathbb{E}_{\mathcal{D}_0}\left[((x \cdot w - y)x \cdot (w - w'))^2\right]$$

$$= \frac{1}{|S_j|} \mathbb{E}_{\mathcal{D}_0}\left[((x \cdot (w - w'))^2 + x \cdot (w' - w_0)x \cdot (w - w') + (w_0 \cdot x - y)x \cdot (w - w'))^2\right]$$

$$\overset{(a)}{\leq} \frac{3}{|S_j|}\left(\mathbb{E}_{\mathcal{D}_0}\left[((w - w') \cdot x)^4\right] + \mathbb{E}_{\mathcal{D}_0}\left[(x \cdot (w' - w_0)x \cdot (w - w'))^2\right]\right. \tag{8}$$

$$\left. + \mathbb{E}_{\mathcal{D}_0}\left[((w_0 \cdot x - y)x \cdot (w - w'))^2\right]\right)$$

$$\overset{(b)}{\leq} \frac{3}{|S_j|}\left(C\,\mathbb{E}_{\mathcal{D}_0}\left[(x \cdot (w - w'))^2\right]^2 + \left(C\,\mathbb{E}_{\mathcal{D}_0}[(x \cdot (w' - w_0))^2] + \sigma^2\right)\mathbb{E}_{\mathcal{D}_0}[(x \cdot (w - w'))^2]\right)$$

$$\overset{(c)}{\leq} \frac{3}{|S_j|}\left(\sqrt{C}\,\mathbb{E}_{\mathcal{D}_0}\left[(x \cdot (w - w'))^2\right] + \left(\sqrt{C\,\mathbb{E}_{\mathcal{D}_0}[(x \cdot (w' - w_0))^2]} + \sigma\right)\sqrt{\mathbb{E}_{\mathcal{D}_0}[(x \cdot (w - w'))^2]}\right)^2.$$

In (a), we use the fact that for any $t, u, v \in \mathbb{R}$, $(t + u + v)^2 \leq 3t^2 + 3u^2 + 3v^2$. In (b) the first term is bounded using assumption 1a, the bound on the second term can be obtained similarly to Equation (5), and the bound on the last term is from Equation (6). Finally, in (c) we use the fact that for any $t, u, v \geq 0$, $(t + u + v)^2 \leq t^2 + u^2 + v^2$.

Using Equation (2) and the equation above, we get

$$\mathbb{E}[a_j] - 2\sqrt{\text{Var}(a_j)} \geq \mathbb{E}_{\mathcal{D}_0}[(x \cdot (w - w'))^2] + \mathbb{E}_{\mathcal{D}_0}[x \cdot (w' - w_0)x \cdot (w - w')]$$

$$- \frac{2\sqrt{3}}{\sqrt{|S_j|}}\left(\sqrt{C}\,\mathbb{E}_{\mathcal{D}_0}\left[(x \cdot (w - w'))^2\right] + \left(\sqrt{C\,\mathbb{E}_{\mathcal{D}_0}[(x \cdot (w' - w_0))^2]} + \sigma\right)\sqrt{\mathbb{E}_{\mathcal{D}_0}[(x \cdot (w - w'))^2]}\right)$$

$$\overset{(a)}{\geq} \left(1 - \frac{\sqrt{12C}}{\sqrt{|S_j|}}\right)\mathbb{E}_{\mathcal{D}_0}[(x \cdot (w - w'))^2] - \sqrt{\mathbb{E}_{\mathcal{D}_0}[(x \cdot (w' - w_0))^2]}\sqrt{\mathbb{E}_{\mathcal{D}_0}[(x \cdot (w - w'))^2]}$$

$$- \frac{\sqrt{12}}{\sqrt{|S_j|}}\left(\left(\sqrt{C\,\mathbb{E}_{\mathcal{D}_0}[(x \cdot (w' - w_0))^2]} + \sigma\right)\sqrt{\mathbb{E}_{\mathcal{D}_0}[(x \cdot (w - w'))^2]}\right)$$

$$\overset{(b)}{\geq} \left(\frac{1}{2}\sqrt{\mathbb{E}_{\mathcal{D}_0}[(x \cdot (w - w'))^2]} - \frac{3}{2}\sqrt{\mathbb{E}_{\mathcal{D}_0}[(x \cdot (w' - w_0))^2]} - \frac{\sqrt{12}}{\sqrt{|S_j|}}\sigma\right)\sqrt{\mathbb{E}_{\mathcal{D}_0}[(x \cdot (w - w'))^2]}$$

$$\overset{(c)}{\geq} \left(\frac{1}{2}\sqrt{\mathbb{E}_{\mathcal{D}_0}[(x \cdot (w - w'))^2]} - \frac{3}{2}\sqrt{\mathbb{E}_{\mathcal{D}_0}[(x \cdot (w' - w_0))^2]} - \sqrt{C_1}\beta\right)\sqrt{\mathbb{E}_{\mathcal{D}_0}[(x \cdot (w - w'))^2]},$$

here, in (a) we use the Cauchy-Schwarz inequality, (b) follows from $|S_j| \geq 48C$, and (c) utilizes $|S_j| \geq \frac{12\sigma^2}{C_1\beta^2}$. Next, we have:

$$\frac{1}{2}\sqrt{\mathbb{E}_{\mathcal{D}_0}[(x \cdot (w - w'))^2]} - \frac{3}{2}\sqrt{\mathbb{E}_{\mathcal{D}_0}[(x \cdot (w' - w_0))^2]} - \sqrt{C_1}\beta \tag{9}$$

$$\overset{(a)}{\geq} \frac{1}{2}\|w - w'\| - \frac{3}{2}\sqrt{C_1}\|w' - w_0\| - \sqrt{C_1}\beta$$

$$\overset{(b)}{\geq} \frac{1}{2}\|w - w'\| - \frac{5}{2}\sqrt{C_1}\beta$$

$$\overset{(c)}{>} \frac{1}{4}\|w - w'\|, \tag{10}$$

here in (a), we use assumption 1b, (b) relies on $w' = w^*$, which implies $\|w' - w_0\| \leq \beta$, and (c) uses the fact that for any $w$ and $w'$ in the while loop of the algorithm, $\|w - w'\| \geq 12C_1\beta$ and $C_1 \geq 1$.

Combining the above two equations, we obtain

$$\mathbb{E}[a_j] - 2\sqrt{\text{Var}(a_j)} > \frac{1}{4}\|w - w'\|^2.$$

Then from Equation (4) it follows that each $a_j > \frac{\|w - w'\|^2}{4}$ with probability $\geq 3/4$. Hence, with probability $\geq 1 - \delta/|L|$ the median of $a_j$ is $> \frac{\|w - w'\|^2}{4}$. $\qquad\square$

# C   Properties of Clipped Gradients

The norm of the expected value and covariance of unclipped gradients for components other than $\mathcal{D}_0$ can be significantly larger than $\mathcal{D}_0$, acting as noise in the recovery process of $\mathcal{D}_0$. When using unclipped gradients, the algorithm's batch size and the number of batches must increase to limit the effect of these noisy components. And while the norm of the expected value and covariance of the unclipped gradient for $\mathcal{D}_0$ follows desired bounds, the maximum value of the unclipped gradient is unbounded, posing difficulties in applying concentration bounds. The following lemma shows that the clipping operation described in the main paper is able to address these challenges.

**Lemma C.1.** *Let $S$ be a collection of random samples drawn from distribution $\mathcal{D}_i$ for some $i \in \{0, 1, ..., k-1\}$. For any $\kappa \geq 0$ and $w \in \mathbb{R}^d$, the clipped gradient satisfies the following properties:*

1. $\|\mathbb{E}[\nabla f(S, w, \kappa)]\| \leq \kappa\sqrt{C_1}$,

2. $\|Cov(\nabla f(S, w, \kappa))\| \leq \frac{1}{|S|}\kappa^2 C_1$,

3. $\|\nabla f(S, w, \kappa)\| \leq \kappa C_2\sqrt{d}$ *almost surely,*

4. $\mathbb{E}\big[\|\nabla f(S, w, \kappa)\|^2\big] \leq C_1\kappa^2 d$,

5. *for all unit vectors $u$,* $\big\|\mathbb{E}\big[(\nabla f(S, w, \kappa) \cdot u)^2\big]\big\| \leq \kappa^2 C_1$.

This lemma implies that for smaller values of $\kappa$, the norm of the expectations and covariance of clipped gradients is bounded by a smaller upper limit. The proof of the lemma is presented in Subsection C.1.

The following theorem demonstrates that by appropriately choosing a sufficiently large value of $\kappa$, the norm of the expected difference between the clipped and unclipped gradients for distribution $\mathcal{D}_0$ can be small:

**Theorem C.2.** *For any $\epsilon > 0$, $\kappa^2 \geq 8CC_1\mathbb{E}_{\mathcal{D}_0}[(y - x \cdot w)^2]/\epsilon$ the norm of difference between expected clipped gradient $\mathbb{E}_{\mathcal{D}_0}[(\nabla f(x, y, w, \kappa)]$ and expected unclipped gradient $\mathbb{E}_{\mathcal{D}_0}[(w \cdot x - y)x]$ is at most,*

$$\|\mathbb{E}_{\mathcal{D}_0}[(\nabla f(x, y, w, \kappa)] - \mathbb{E}_{\mathcal{D}_0}[(w \cdot x - y)x]\| \leq \epsilon\|w - w_0\|,$$

*where $\mathbb{E}_{\mathcal{D}_0}[(w \cdot x - y)x] = \Sigma_0(w - w_0)$.*

The theorem shows in order to estimate the expectation of gradients at point $w$ for distribution $\mathcal{D}_0$, it is sufficient to estimate the expectation of clipped gradients at point $w$, as long as the clipping parameter $\kappa$ is chosen to be at least $\Omega\left(\sqrt{\frac{\mathbb{E}_{\mathcal{D}_0}[(y - x \cdot w)^2]}{\epsilon}}\right)$.

Intuitively, when $\kappa$ is much larger than $\mathbb{E}_{\mathcal{D}_0}[|y - x \cdot w|]$, with high probability the clipped and unclipped gradients at point $w$ for a random sample from $\mathcal{D}_0$ will be identical. The proof of the theorem is a bit more nuanced and involves leveraging the symmetry of noise distribution and $L4 - L2$ hypercontractivity of distribution of $x$. The proof appears in Subsection C.2.

In the algorithm, we set $\kappa$ to approximately $\Theta(\sqrt{(\mathbb{E}_{\mathcal{D}_0}[(y - x \cdot w)^2] + \sigma^2)/\epsilon})$. This choice ensures that $\kappa$ is close to the minimum value recommended by Theorem C.2 for preserving the gradient expectation of $\mathcal{D}_0$. By selecting a small $\kappa$, we ensure a tighter upper bound on the expectation and covariance of the clipped gradient for other components, as described in Lemma C.1. The use of the clipping operation also assists in obtaining improved bounds on the tails of the gradient by limiting the maximum possible norm of the gradients after clipping, as stated in item 3 of the lemma.

## C.1   Proof of Lemma C.1

*Proof.* Since $\nabla f(S, w, \kappa)$ is average of clipped gradients of $|S|$ independent samples from $\mathcal{D}_i$, it follows that

a) $\mathbb{E}[\nabla f(S, w, \kappa)] = \mathbb{E}_{\mathcal{D}_i}[\nabla f(x, y, w, \kappa)]$,

b) $Cov(\nabla f(S, w, \kappa)) = \frac{1}{|S|}Cov_{\mathcal{D}_i}(\nabla f(x, y, w, \kappa))$,

c) $\|\nabla f(S, w, \kappa)\| \leq \operatorname{ess\,sup}_{(x,y) \sim \mathcal{D}_i} \|\nabla f(x, y, w, \kappa)\|$ a.s.,

d) $\mathbb{E}\big[\|\nabla f(S, w, \kappa)\|^2\big] \leq \mathbb{E}_{\mathcal{D}_i}\big[\|\nabla f(x, y, w, \kappa)\|^2\big]$, and

e) for all vectors $u$, $\big\|\mathbb{E}\big[(\nabla f(S, w, \kappa) \cdot u)^2\big]\big\| \leq \big\|\mathbb{E}_{\mathcal{D}_i}\big[(\nabla f(x, y, w, \kappa) \cdot u)^2\big]\big\|$.

We will now proceed to prove the five claims in the lemma by using these properties.

Firstly, we can analyze the expected norm of $\mathbb{E}_{\mathcal{D}_i}[\nabla f(x, y, w, \kappa)]$ as follows:

$$
\begin{aligned}
\|\mathbb{E}_{\mathcal{D}_i}[\nabla f(x, y, w, \kappa)]\| &= \max_{\|u\|} \|\mathbb{E}_{\mathcal{D}_i}[(\nabla f(x, y, w, \kappa) \cdot u)]\| \\
&\leq \max_{\|u\|} \|\mathbb{E}_{\mathcal{D}_i}[(\nabla f(x, y, w, \kappa) \cdot u)^2]^{1/2}\| \\
&\leq \max_{\|u\|} \|\mathbb{E}_{\mathcal{D}_i}[(\kappa x \cdot u)^2]^{1/2}\| \leq \kappa \sqrt{C_1},
\end{aligned}
$$

here the first inequality follows from the Cauchy–Schwarz inequality and the last inequality follows from assumptions on distributions $\mathcal{D}_i$. Combining the above inequality with item a) above proves the first claim in the lemma.

Next, to prove the second claim in the lemma, we first establish bounds for the norm of the covariance of the clipped gradient of a random sample:

$$
\begin{aligned}
\|\operatorname{Cov}_{\mathcal{D}_i}(\nabla f(x, y, w, \kappa))\| &= \max_{\|u\|} \operatorname{Var}_{\mathcal{D}_i}(\nabla f(x, y, w, \kappa) \cdot u) \\
&\leq \max_{\|u\|} \mathbb{E}_{\mathcal{D}_i}[(\nabla f(x, y, w, \kappa) \cdot u)^2] \\
&\leq \max_{\|u\|} \mathbb{E}_{\mathcal{D}_i}[(\kappa x \cdot u)^2] \leq \kappa^2 C_1.
\end{aligned}
$$

By using the above bound and combining it with item b), we establish the second claim in the lemma.

To prove the third item in the lemma, we first bound the norm of the clipped gradient:

$$
\operatorname*{ess\,sup}_{(x,y) \sim \mathcal{D}_i} \|\nabla f(x, y, w, \kappa)\| \leq \kappa \operatorname*{ess\,sup}_{(x,y) \sim \mathcal{D}_i} \|x\| \leq \kappa C_2 \sqrt{d}.
$$

We then combine this bound with item c) to prove the third claim in the lemma.

Next, we bound the expected value of the square of the norm of the clipped gradient of a random sample,

$$
\mathbb{E}_{\mathcal{D}_i}\big[\|\nabla f(x, y, w, \kappa)\|^2\big] \leq \mathbb{E}_{\mathcal{D}_i}\big[\kappa^2 \|x\|^2\big] = \kappa^2 \operatorname{Tr}(\Sigma_i) \leq \kappa^2 d \|\Sigma_i\| \leq C_1 \kappa^2 d.
$$

This bound, combined with item d), proves the fourth claim in the lemma.

Finally, for any unit vector $u$, we bound

$$
\big\|\mathbb{E}_{\mathcal{D}_i}\big[(\nabla f(x, y, w, \kappa) \cdot u)^2\big]\big\| \leq \kappa^2 \big\|\mathbb{E}_{\mathcal{D}_i}\big[(x \cdot u)^2\big]\big\| \leq \kappa^2 \|\Sigma_i\| \leq \kappa^2 C_1.
$$

This bound, combined with item e), shows the fifth claim in the lemma. $\qquad\square$

## C.2 Proof of Theorem C.2

We will utilize the following auxiliary lemma in the proof of the theorem. This lemma applies to general random variables.

**Lemma C.3.** *For any $a \in \mathbb{R}$, $b > 0$ and a symmetric random variable $z$,*

$$
\left| \mathbb{E}\left[ (a + z) - \frac{(a + z)b}{\max(|a + z|, b)} \right] \right| \leq 2|a| \Pr(z > b - |a|)
$$

*Proof.* We assume $a \geq 0$ and prove the lemma for this case. The statement for $a < 0$ case then follows from symmetry.

We rewrite the term inside the expectation in terms of indicator random variables:

$$(a + z) - \frac{(a + z)b}{\max(|a + z|, b)}$$
$$= (a + z - b) \cdot \mathbb{1}(z > b - a) + (a + z + b) \cdot \mathbb{1}(z < -b - a)$$
$$= (a + z - b) \cdot \mathbb{1}(b - a < z \leq b + a) + (a + z - b) \cdot \mathbb{1}(z > b + a) + (a + z + b) \cdot \mathbb{1}(z < -b - a).$$

Taking the expectation on both sides,

$$\mathbb{E}\left[(a + z) - \frac{(a + z)b}{\max(|a + z|, b)}\right]$$
$$= \mathbb{E}[(a + z - b) \cdot \mathbb{1}(b - a < z \leq b + a)] + \mathbb{E}[(a + z - b) \cdot \mathbb{1}(z > b + a)]$$
$$+ \mathbb{E}[(a + z + b) \cdot \mathbb{1}(z < -b - a)]$$
$$= \mathbb{E}[(a + z - b) \cdot \mathbb{1}(b - a < z \leq b + a)] + 2a \Pr(z > b + a),$$

where the last step follows because $z$ is symmetric.

Next,

$$\left|\mathbb{E}\left[(a + z) - \frac{(a + z)b}{\max(|a + z|, b)}\right]\right| = \mathbb{E}[|a + z - b| \cdot \mathbb{1}(b - a < z \leq b + a)] + 2|a| \Pr(z > b + a)$$
$$\leq \mathbb{E}[|2a| \cdot \mathbb{1}(b - a < z \leq b + a)] + 2|a| \Pr(z > b + a)$$
$$\leq 2|a| \Pr(b - a < z \leq b + a) + 2|a| \Pr(z > b + a)$$
$$= 2|a| \Pr(z > b - a).$$

$\square$

Next, we proceed with the proof of Theorem C.2 using the aforementioned lemma.

*Proof of Theorem C.2.* Let $(x, y)$ be a random sample from distribution $\mathcal{D}_0$, and let $\eta = y - w_0 \cdot x$ denote the noise. Recall that $\eta$ is independent of $x$.

Note that:

$$(w \cdot x - y)x = ((w - w_0) \cdot x - \eta)x.$$

We will now evaluate the expected value of the unclipped gradient.

$$\mathbb{E}_{\mathcal{D}_0}[(w \cdot x - y)x] = \mathbb{E}_{\mathcal{D}_0}[((w - w_0) \cdot x - \eta)x] = \mathbb{E}_{\mathcal{D}_0}[((w - w_0) \cdot x)x] = \Sigma_0(w - w_0). \quad (11)$$

Next, we will bound the norm of the expected value of $(w \cdot x - y)x - \nabla f(x, y, w, \kappa)$, which represents the difference between the clipped gradient and the true gradient. We first expand this expression:

$$(w \cdot x - y)x - \nabla f(x, y, w, \kappa) = \left((w \cdot x - y) - \frac{(w \cdot x - y)}{|w \cdot x - y| \vee \kappa}\kappa\right)x$$
$$= \left(((w - w_0) \cdot x - \eta) - \frac{((w - w_0) \cdot x - \eta)}{|(w - w_0) \cdot x - \eta| \vee \kappa}\kappa\right)x. \quad (12)$$

Next, by applying Lemma C.3, we have

$$\mathbb{E}_\eta\left[((w - w_0) \cdot x - \eta) - \frac{((w - w_0) \cdot x - \eta)}{|(w - w_0) \cdot x - \eta| \vee \kappa}\kappa\right]$$
$$\leq 2|(w - w_0) \cdot x| \cdot \Pr(\eta > \kappa - |(w - w_0) \cdot x|).$$

Note that in the above expectation, we fixed $x$ and took the expectation over noise $\eta$.

Let $Z := \mathbb{1}(|x \cdot (w - w_0)| \geq \kappa/2)$. Observe that $\Pr(\eta > \kappa - |(w - w_0) \cdot x|) \leq Z + \Pr(\eta > \kappa/2)$. Combining this observation with the above equation, we have:

$$\mathbb{E}_\eta\left[((w - w_0) \cdot x - \eta) - \frac{((w - w_0) \cdot x - \eta)}{|(w - w_0) \cdot x - \eta| \vee \kappa}\kappa\right] \leq 2|(w - w_0) \cdot x| \cdot (\Pr(\eta > \kappa/2) + Z).$$
$$(13)$$

Then, for any unit vector $v \in \mathbb{R}^d$, we have

$$|\mathbb{E}_{\mathcal{D}_0}[((w \cdot x - y)x - \nabla f(x, y, w, \kappa)) \cdot v]|$$
$$= |\mathbb{E}_{x \sim \mathcal{D}_0}[\mathbb{E}_{\eta \sim \mathcal{D}_0}[((w \cdot x - y)x - \nabla f(x, y, w, \kappa)) \cdot v]]|$$
$$\le \mathbb{E}_{x \sim \mathcal{D}_0}[|\mathbb{E}_{\eta \sim \mathcal{D}_0}[((w \cdot x - y)x - \nabla f(x, y, w, \kappa)) \cdot v]|]$$
$$\le \mathbb{E}_{x \sim \mathcal{D}_0}[2|(w - w_0) \cdot x| \cdot |x \cdot v|(Z + \Pr(\eta > \kappa/2))]$$
$$\le 2\,\mathbb{E}_{\mathcal{D}_0}[Z \cdot |(w - w_0) \cdot x| \cdot |x \cdot v|] + 2\Pr(\eta > \kappa/2)\,\mathbb{E}_{\mathcal{D}_0}[|(w - w_0) \cdot x| \cdot |x \cdot v|], \qquad (14)$$

here the second last inequality follows from Equation (12) and Equation (13). Next, we bound the two terms on the right one by one. We start with the first term:

$$\mathbb{E}_{\mathcal{D}_0}[Z \cdot |(x \cdot (w - w_0))(x \cdot v)|] \overset{(a)}{\le} \left(\mathbb{E}[(Z)^2] \cdot \mathbb{E}_{\mathcal{D}_0}[(x \cdot (w - w_0))^2(x \cdot v)^2]\right)^{1/2}$$

$$\overset{(b)}{\le} \left(\mathbb{E}[Z] \cdot \mathbb{E}_{\mathcal{D}_0}[(x \cdot (w - w_0))^4]^{1/2}\,\mathbb{E}_{\mathcal{D}_0}[(x \cdot v)^4]^{1/2}\right)^{1/2}$$

$$\overset{(c)}{\le} \left(\mathbb{E}[Z] \cdot C\,\mathbb{E}_{\mathcal{D}_0}[(x \cdot (w - w_0))^2]\,\mathbb{E}_{\mathcal{D}_0}[(x \cdot v)^2]\right)^{1/2}$$

$$\overset{(d)}{\le} \left(CC_1 \Pr[|x \cdot (w - w_0)| \ge \kappa/2] \cdot \mathbb{E}_{\mathcal{D}_0}[(x \cdot (w - w_0))^2]\right)^{1/2},$$
$$(15)$$

where (a) used the Cauchy-Schwarz inequality, (b) used the fact that $Z$ is an indicator random variable, hence, $Z^2 = Z$, and the Cauchy-Schwarz inequality, (c) uses $L4 - L2$ hypercontractivity, and (d) follows from the definition of $Z$ and the assumption that $\|\Sigma_0\| \le C_1$.

Similarly, we can show that

$$\mathbb{E}_{\mathcal{D}_0}[|(w - w_0) \cdot x| \cdot |x \cdot v|] \le \left(C_1\,\mathbb{E}_{\mathcal{D}_0}[(x \cdot (w - w_0))^2]\right)^{1/2}. \qquad (16)$$

Applying the Markov inequality to $\eta^2$ we get:

$$\Pr[|\eta| \ge \kappa/2] \le \frac{\mathbb{E}_{\mathcal{D}_0}[\eta^2]}{(\kappa/2)^2}. \qquad (17)$$

Similarly, applying the Markov inequality to $|x \cdot (w - w_0)|^4$ yields:

$$\Pr[|x \cdot (w - w_0)| \ge \kappa/2] \le \frac{\mathbb{E}_{\mathcal{D}_0}[|x \cdot (w - w_0)|^4]}{(\kappa/2)^4} \le \frac{C\,\mathbb{E}_{\mathcal{D}_0}[|x \cdot (w - w_0)|^2]^2}{(\kappa/2)^4}, \qquad (18)$$

where the last inequality uses $L4 - L2$ hypercontractivity.

Combining Equations (14), (15), (16), (17) and (18), we have

$$|\mathbb{E}_{\mathcal{D}_0}[((w \cdot x - y)x - \nabla f(x, y, w, \kappa)) \cdot v]|$$

$$\le \frac{8C\sqrt{C_1}\left(\mathbb{E}_{\mathcal{D}_0}[(x \cdot (w - w_0))^2]\right)^{3/2}}{\kappa^2} + \frac{8\sqrt{C_1}\,\mathbb{E}_{\mathcal{D}_0}[\eta^2]\left(\mathbb{E}_{\mathcal{D}_0}[(x \cdot (w - w_0))^2]\right)^{1/2}}{\kappa^2}$$

$$\le \frac{8C\sqrt{C_1}\left(\mathbb{E}_{\mathcal{D}_0}[(x \cdot (w - w_0))^2]\right)^{1/2}\left(\left(\mathbb{E}_{\mathcal{D}_0}[(x \cdot (w - w_0))^2]\right) + \mathbb{E}_{\mathcal{D}_0}[\eta^2]/C\right)}{\kappa^2}$$

$$\overset{(a)}{\le} \frac{\epsilon\left(\mathbb{E}_{\mathcal{D}_0}[(x \cdot (w - w_0))^2]\right)^{1/2}}{\sqrt{C_1}} \cdot \frac{\left(\left(\mathbb{E}_{\mathcal{D}_0}[(x \cdot (w - w_0))^2]\right) + \mathbb{E}_{\mathcal{D}_0}[\eta^2]/C\right)}{\mathbb{E}_{\mathcal{D}_0}[(y - x \cdot w)^2]}$$

$$\overset{(b)}{\le} \epsilon\|w - w_0\| \cdot \frac{\left(\left(\mathbb{E}_{\mathcal{D}_0}[(x \cdot (w - w_0))^2]\right) + \mathbb{E}_{\mathcal{D}_0}[\eta^2]\right)}{\mathbb{E}_{\mathcal{D}_0}[(y - x \cdot w)^2]}$$

$$\overset{(c)}{=} \epsilon\|w - w_0\|,$$

here inequality (a) follows from the lower bound on $\kappa^2$ in theorem, inequality (b) follows from Assumption 1b and $C \ge 1$, and the last equality follows since $y - x \cdot w = x(w - w_0) + \eta$, and $x$ and $\eta$ are independent.

Note that the above bound holds for all unit vectors $v$, therefore,

$$\|\mathbb{E}_{\mathcal{D}_0}[((w \cdot x - y)x - \nabla f(x, y, w, \kappa))]\| \le \max_{\|v\|} \mathbb{E}_{\mathcal{D}_0}[((w \cdot x - y)x - \nabla f(x, y, w, \kappa))] \cdot v \le \epsilon\|w - w_0\|.$$

$$\square$$

# D Estimation of clipping parameter

---

**Algorithm 4** CLIPEST

---

1: **Input:** A collection of samples $S^*$ from $\mathcal{D}_0$, $w$, $\epsilon$, $\delta'$, $\sigma$, $C$, and $C_1$.
2: **Output:** clipping parameter $\kappa$
3: $T \leftarrow \Theta(\log 1/\delta')$
4: Divide $S^*$ into $T$ equal parts randomly, and denote them as $\{S_j^*\}_{j \in [T]}$
5: $\theta \leftarrow \text{Median}\left\{ \frac{1}{|S_j^*|} \sum_{(x,y) \in S_j^*} (x \cdot w - y)^2 : j \in [T] \right\}$
6: $\kappa \leftarrow \sqrt{\frac{32(C+1)C_1(\theta + 17\sigma^2)}{\epsilon}}$
7: Return $\kappa$

---

In round $r$, to set $\kappa \approx \sqrt{\mathbb{E}_{\mathcal{D}_0}[(y - x \cdot w)^2]/\epsilon}$ at point $w = \hat{w}^{(r)}$, the main algorithm 1 runs subroutine CLIPEST 4 for $S^* = S_1^{b^*,(r)}$ and $w = \hat{w}^{(r)}$. Recall that $S_1^{b^*,(r)}$ is collection of i.i.d. samples from $\mathcal{D}_0$. Using these samples this subroutine estimates $\mathbb{E}_{\mathcal{D}_0}[(y - x \cdot w)^2]$ at $w = \hat{w}^{(r)}$ using the median of means and then use it to obtain $\kappa$ in the desired range. The following theorem provides the guarantees on the estimation of $\kappa$ by this subroutine.

**Theorem D.1.** *For $\epsilon > 0$, $T \geq \Omega(\log 1/\delta')$ and $|S^*| \geq 64C^2 T$ and $w \in \mathbb{R}^d$. With probability $\geq 1 - \delta'$, the clipping parameter $\kappa$ returned by subroutine CLIPEST satisfy,*

$$\sqrt{\frac{8(C+1)C_1 \cdot \mathbb{E}_{\mathcal{D}_0}[(y - x \cdot w)^2]}{\epsilon}} \leq \kappa \leq 28 \sqrt{\frac{2(C+1)C_1 \left( \mathbb{E}_{\mathcal{D}_0}[((w - w_0) \cdot x)^2] + \sigma^2 \right)}{\epsilon}}.$$

To prove the theorem, we will make use of the following lemma:

**Lemma D.2.** *Let $S$ be a collection of $m \geq 64C^2$ i.i.d. samples from $\mathcal{D}_0$ and $w \in \mathbb{R}^d$, then with probability at least $7/8$, the following holds:*

$$\frac{1}{4} \mathbb{E}_{\mathcal{D}_0}[(w \cdot x - y)^2] - 17\sigma^2 \leq \frac{1}{m} \sum_{(x,y) \in S} (y - w \cdot x)^2 \leq 3 \mathbb{E}_{\mathcal{D}_0}[((w - w_0) \cdot x)^2] + 32\sigma^2.$$

*Proof.* We start by expanding the expression:

$$\frac{1}{m} \sum_{(x,y) \in S} (w \cdot x - y)^2 = \frac{1}{m} \sum_{(x,y) \in S} ((w - w_0) \cdot x + (w_0 \cdot x - y))^2$$

$$= \frac{1}{m} \sum_{(x,y) \in S} \left( ((w - w_0) \cdot x)^2 + 2((w - w_0) \cdot x)(w_0 \cdot x - y) + (w_0 \cdot x - y)^2 \right)$$

$$\geq \frac{1}{m} \sum_{(x,y) \in S} \left( \frac{1}{2}((w - w_0) \cdot x)^2 - (w_0 \cdot x - y)^2 \right), \tag{19}$$

where the last inequality follows since for any $a, b$, we have $a^2 + 2ab + b^2 \geq a^2/2 - b^2$.

Similarly, we can show:

$$\frac{1}{m} \sum_{(x,y) \in S} (w \cdot x - y)^2 \leq \frac{1}{m} \sum_{(x,y) \in S} \left( 2((w - w_0) \cdot x)^2 + 2(w_0 \cdot x - y)^2 \right). \tag{20}$$

Since $S$ contains independent samples from $\mathcal{D}_0$, we have:

$$\mathbb{E}\left[ \frac{1}{m} \sum_{(x,y) \in S} (((w - w_0) \cdot x)^2 \right] = \mathbb{E}_{\mathcal{D}_0}[((w - w_0) \cdot x)^2],$$

and

$$\text{Var}\left( \frac{1}{m} \sum_{(x,y) \in S} ((w - w_0) \cdot x)^2 \right) = \frac{\text{Var}_{\mathcal{D}_0}(((w - w_0) \cdot x)^2)}{m}$$

$$\leq \frac{\mathbb{E}_{\mathcal{D}_0}[((w - w_0) \cdot x)^4]}{m} \leq \frac{C \mathbb{E}_{\mathcal{D}_0}[((w - w_0) \cdot x)^2]^2}{m},$$

where the last inequality follows from $L4$-$L2$ hypercontractivity.

For any $a > 0$, using Chebyshev's inequality,

$$\Pr\left[\left\|\frac{1}{m}\sum_{(x,y)\in S}((w-w_0)\cdot x)^2 - \mathbb{E}_{\mathcal{D}_0}[((w-w_0)\cdot x)^2]\right\| \geq a\frac{C\,\mathbb{E}_{\mathcal{D}_0}[((w-w_0)\cdot x)^2]}{\sqrt{m}}\right] \leq \frac{1}{a^2}. \tag{21}$$

Using the Markov inequality, for any $a > 0$, we have:

$$\Pr\left[\frac{1}{m}\sum_{(x,y)\in S}(w_0\cdot x-y)^2 > a^2\sigma^2\right] \leq \frac{\mathbb{E}_{\mathcal{D}_0}[(w_0\cdot x-y)^2]}{a^2\sigma^2} \leq \frac{1}{a^2}. \tag{22}$$

By combining the equations above, we can derive the following inequality:

With probability $\geq 1 - \frac{2}{a^2}$, the following holds:

$$\frac{1}{m}\sum_{(x,y)\in S}(w\cdot x-y)^2 \geq \frac{1}{2}\mathbb{E}_{\mathcal{D}_0}[((w-w_0)\cdot x)^2](1-a\frac{C}{\sqrt{m}}) - a^2\sigma^2,$$

and

$$\frac{1}{m}\sum_{(x,y)\in S}(w\cdot x-y)^2 \leq 2\,\mathbb{E}_{\mathcal{D}_0}[((w-w_0)\cdot x)^2](1+a\frac{C}{\sqrt{m}}) + 2a^2\sigma^2.$$

By choosing $a = 4$ and using $m \geq 64C^2$ in the above equation, we can conclude that with probability $\geq 1 - \frac{2}{a^2}$, the following holds:

$$\frac{1}{4}\mathbb{E}_{\mathcal{D}_0}[((w-w_0)\cdot x)^2] - 16\sigma^2 \leq \frac{1}{m}\sum_{(x,y)\in S}(w\cdot x-y)^2 \leq 3\,\mathbb{E}_{\mathcal{D}_0}[((w-w_0)\cdot x)^2] + 32\sigma^2.$$

Next, note that

$$\begin{aligned}
\mathbb{E}_{\mathcal{D}_0}[(w\cdot x-y)^2] &= \mathbb{E}_{\mathcal{D}_0}[((w-w_0)\cdot x - (y-w_0\cdot x))^2]\\
&\overset{(a)}{=} \mathbb{E}_{\mathcal{D}_0}[((w-w_0)\cdot x)^2] + \mathbb{E}_{\mathcal{D}_0}[(y-w_0\cdot x)^2]\\
&\overset{(b)}{\leq} \mathbb{E}_{\mathcal{D}_0}[((w-w_0)\cdot x)^2] + \sigma^2,
\end{aligned}$$

here (a) follows since $x$ is independent of the output noise $y - w_0\cdot x$, and (b) follows since the output noise $y - w_0\cdot x$ is zero mean and has a variance at most $\sigma^2$. Combining the above two equations completes the proof. $\qquad\square$

Now we prove Theorem D.1 using the above lemma:

*Proof of Theorem D.1.* From the previous lemma and Chernoff bound it follows that with probability $\geq 1 - \delta'$,

$$\frac{1}{4}\mathbb{E}_{\mathcal{D}_0}[(w\cdot x-y)^2] - 17\sigma^2 \leq \theta \leq 3\,\mathbb{E}_{\mathcal{D}_0}[((w-w_0)\cdot x)^2] + 32\sigma^2.$$

Then bound on $\kappa$ follows from the relation $\kappa = \sqrt{\frac{32(C+1)C_1(\theta+17\sigma^2)}{\epsilon}}$. $\qquad\square$

# E   Subspace Estimation

As a part of gradient estimation in step $r$, the main algorithm 1 uses subroutine GRADSUBEST for $\widehat{B} = B_s^{(r)}$ and $w = \hat{w}^{(r)}$. Recall that $B_s^{(r)}$ is a random subset of the collection of small batches $B_s$.

The purpose of this subroutine is to estimate a smaller subspace of $\mathbb{R}^d$ such that for distribution $\mathcal{D}_0$, the expectation of the projection of the clipped gradient onto this subspace closely approximates the

---

**Algorithm 5** GRADSUBEST

---

1: **Input:** A collection of medium batches $\widehat{B}$, $\kappa$, $w$, $\ell$
2: **Output:** A rank $\ell$ projection matrix.
3: For each $b \in \widehat{B}$ divide its samples $S^b$ into two equal random parts $S_1^b$ and $S_2^b$
4: $A \leftarrow \frac{1}{2|\widehat{B}|} \sum_{b \in \widehat{B}} \left( \nabla f(S_1^b, w, \kappa) \nabla f(S_2^b, w, \kappa)^\intercal + \nabla f(S_2^b, w, \kappa) \nabla f(S_1^b, w, \kappa)^\intercal \right)$
5: $U \leftarrow [u_1, u_2, ..., u_\ell]$, where $\{u_i\}$'s are top $\ell$ singular vectors of $A$
6: Return $UU^\intercal$

---

true expectation of the clipped gradient, for distribution $\mathcal{D}_0$. This reduction to a smaller subspace helps reduce the number of medium-sized batches and their required length in the subsequent part of the algorithm.

The following theorem characterizes the final guarantee for subroutine GRADSUBEST.

**Theorem E.1.** *Let $p_0$ denote the fraction of batches in $\widehat{B}$ that are sampled from $\mathcal{D}_0$. For any $\epsilon, \delta' > 0$, and $\widehat{B} = \Omega\left( \frac{d}{\alpha_s \epsilon^2} \left( \frac{1}{\alpha_s \epsilon^2} + \frac{C_2^2}{C_1} \right) \log \frac{d}{\delta'} \right)$, $p_0 \geq \alpha_s/2$ and $\ell \geq \min\{k, \frac{1}{2\alpha_s \epsilon^2}\}$, with probability $\geq 1 - \delta'$, the projection matrix $UU^\intercal$ returned by subroutine GRADSUBEST satisfy*

$$\|(I - UU^\intercal) \mathbb{E}_{\mathcal{D}_0}[\nabla f(x, y, w, \kappa)]\| \leq 4\epsilon\kappa\sqrt{C_1}.$$

The above theorem implies that the difference between the expectation of the clipped gradient and the expectation of projection of the clipped gradient for distribution $\mathcal{D}_0$ is small. Next, we present the description of the subroutine GRADSUBEST and provide a brief outline of the proof for the theorem before formally proving it in the subsequent subsection.

The subroutine divides samples in each batch $b \in \widehat{B}$ into two parts, namely $S_1^b$ and $S_2^b$. Then it computes the clipped gradients $u^b := \nabla f(S_1^b, w, \kappa)$ and $v^b := \nabla f(S_2^b, w, \kappa)$. From linearity of expectation, for any $i$ and batch $b$ that contain i.i.d. samples from $\mathcal{D}_i$, $\mathbb{E}[u^b] = \mathbb{E}[v^b] = \mathbb{E}_{\mathcal{D}_i}[\nabla f(x, y, w, \kappa)]$. The subroutine defines $A = \sum_{b \in \widehat{B}} \frac{1}{2|\widehat{B}|} u^b(v^b)^\intercal + v^b(u^b)^\intercal$. Let $p_i$ denote the fraction of batches in $\widehat{B}$ that have samples from $\mathcal{D}_i$. Then using the linearity of expectation, we have:

$$\mathbb{E}[A] = \sum_{i=0}^{k-1} p_i \, \mathbb{E}_{\mathcal{D}_i}[\nabla f(x, y, w, \kappa)] \, \mathbb{E}_{\mathcal{D}_i}[\nabla f(x, y, w, \kappa)]^\intercal.$$

It is evident that if the matrix $U$ is formed by selecting the top $k$ singular vectors of $\mathbb{E}[A]$, then the projection of $\mathbb{E}_{\mathcal{D}_0}[\nabla f(x, y, w, \kappa)]$ onto $UU^\intercal$ corresponds to itself, and the guarantee stated in the theorem holds. However, we do not have access to $\mathbb{E}[A]$, and furthermore, when the number of components $k$ is large, it may be desirable to obtain a subspace of smaller size than $k$.

To address the first challenge, Theorem E.2 in the next subsection shows that $\|A - \mathbb{E}[A]\|$ is small. This theorem permits the usage of $A$ as a substitute for $\mathbb{E}[A]$. The clipping operation, introduced in the previous subsection, plays a crucial role in the proof of Theorem E.2 by controlling the norm of the expectation and the covariance of the clipped gradient for other components, and the maximum length of clipped gradients across all components. This is crucial for obtaining a good bound on the number of small-size batches required. Additionally, the clipping operation ensures that the subroutine remains robust to arbitrary input-output relationships for other components.

Furthermore, the clipping operation assists in addressing the second challenge by ensuring a uniform upper bound on the norm of the expectation of all components, i.e., $\| \mathbb{E}_{\mathcal{D}_i}[\nabla f(x, y, w, \kappa)]\| \leq \mathcal{O}(\kappa)$. Leveraging this property, Lemma E.3 demonstrates that it suffices to estimate the top $\approx 1/p_0$-dimensional subspace. Intuitively, this is because the infinitely many components can create at most approximately $1/p_0$ directions with weights greater than $p_0$, indicating that the direction of $\mathcal{D}_0$ must be present in the top $\Theta(1/p_0)$ subspace.

Since $\widehat{B} = B_s^{(r)}$ is obtained by randomly partitioning $B_s$ into $R$ subsets, and $B_s$ contains a fraction of at least $\alpha_s$ batches with samples from $\mathcal{D}_0$, it holds with high probability that $p_0 \gtrsim \alpha_s$. Consequently, when $\ell \geq \min\{k, \Omega(\frac{1}{\alpha_s})\}$, the subspace corresponding to the top $\ell$ singular vectors of $A$ satisfies the desired property in the Theorem E.1.

We note that the construction of matrix $A$ in subroutine GRADSUBEST is inspired by previous work [KSS+20]. However, while they employed it to approximate the $k$-dimensional subspace of

the true regression vectors for all components, we focus exclusively on one distribution $\mathcal{D}_0$ at a time and recover a subspace such that, for distribution $\mathcal{D}_0$, the expectation of the projection of the clipped gradient on this subspace closely matches the true expectation of the clipped gradient.

It is worth noting that, in addition to repurposing the subroutine from [KSS$^+$20], we achieve four significant improvements:

1) A more meticulous statistical analysis and the use of clipping enable our algorithm to handle heavy-tailed distributions for both noise and input distributions. 2) Clipping also facilitates the inclusion of arbitrary input-output relationships for other components. The next two improvements are attributed to an improved linear algebraic analysis. Specifically, our Lemma E.3 enhances the matrix perturbation bounds found in [LL18] and [KSS$^+$20]. These enhancements enable us to: 3) Provide meaningful guarantees even when the number of components $k$ is very large, 4) reduce the number of batches required when the distance between the regression vectors is small.

## E.1 Proof of Theorem E.1

To prove Theorem E.1, in the following theorem, we will first demonstrate that the term $\|A - \mathbb{E}[A]\|$ is small when given enough batches.

**Theorem E.2.** *For $0 \leq i \leq k-1$, let $z_i = \mathbb{E}_{\mathcal{D}_i}[\nabla f(x, y, w, \kappa)]$, and $p_i$ denote the fraction of batches in $\widehat{B}$ that have samples from $\mathcal{D}_i$. For any $\epsilon, \delta' > 0$, and $\widehat{B} = \Omega\left(\frac{d}{\alpha_s \epsilon^2}\left(\frac{1}{\alpha_s \epsilon^2} + \frac{C_2^2}{C_1}\right) \log \frac{d}{\delta'}\right)$, with probability at least $1 - \delta'$,*

$$\left\| A - \sum_{i=0}^{k-1} p_i z_i z_i^\mathsf{T} \right\| \leq \alpha_s \epsilon^2 \kappa^2 C_1,$$

*where $A$ is the matrix defined in subroutine GRADSUBEST.*

*Proof.* Let $Z^b := \nabla f(S_1^b, w, \kappa) \nabla f(S_2^b, w, \kappa)^\mathsf{T}$.

Note that

$$A = \frac{1}{2|\widehat{B}|} \sum_{b \in \widehat{B}} (Z^b + (Z^b)^\mathsf{T}).$$

Then, from the triangle inequality, we have:

$$\left\| A - \sum_{i=0}^{k-1} p_i z_i z_i^\mathsf{T} \right\| \leq \frac{1}{2}\left\| \frac{1}{|\widehat{B}|} \sum_{b \in \widehat{B}} Z^b - \sum_{i=0}^{k-1} p_i z_i z_i^\mathsf{T} \right\| + \frac{1}{2}\left\| \frac{1}{|\widehat{B}|} \sum_{b \in \widehat{B}} (Z^b)^\mathsf{T} - \sum_{i=0}^{k-1} p_i z_i z_i^\mathsf{T} \right\|$$

$$= \left\| \frac{1}{|\widehat{B}|} \sum_{b \in \widehat{B}} Z^b - \sum_{i=0}^{k-1} p_i z_i z_i^\mathsf{T} \right\|.$$

For a batch $b$ sampled from distribution $\mathcal{D}_i$, we have:

$$\mathbb{E}[Z^b] = \mathbb{E}[\nabla f(S_1^b, w, \kappa) \nabla f(S_2^b, w, \kappa)^\mathsf{T}]$$
$$= \mathbb{E}[\nabla f(S_1^b, w, \kappa)]\, \mathbb{E}[\nabla f(S_2^b, w, \kappa)^\mathsf{T}]$$
$$= \mathbb{E}_{\mathcal{D}_i}[\nabla f(x, y, w, \kappa)]\, \mathbb{E}_{\mathcal{D}_i}[\nabla f(x, y, w, \kappa)^\mathsf{T}] = z_i z_i^\mathsf{T},$$

where the second inequality follows since samples in $S_1^b$ and $S_2^b$ are independent, and the third equality follows from the linearity of expectation.

It follows that

$$\frac{1}{|\widehat{B}|} \sum_{b \in \widehat{B}} \mathbb{E}[Z^b] = \sum_{i=0}^{k-1} p_i z_i z_i^\mathsf{T},$$

and

$$\left\| A - \sum_{i=0}^{k-1} p_i z_i z_i^\intercal \right\| \leq \left\| \frac{1}{|\widehat{B}|} \sum_{b \in \widehat{B}} Z^b - \frac{1}{|\widehat{B}|} \sum_{b \in \widehat{B}} \mathbb{E}[Z^b] \right\|. \tag{23}$$

To complete the proof, we will prove a high probability bound on the term on the right by applying the Matrix Bernstein inequality. To apply this inequality, we first upper bound $|Z^b|$ as follows:

$$\|Z^b\| = \|\nabla f(S_1^b, w, \kappa)\nabla f(S_2^b, w, \kappa)^\intercal\| \leq \|\nabla f(S_1^b, w, \kappa)\| \cdot \|\nabla f(S_2^b, w, \kappa)^\intercal\|.$$

From item 3 in Lemma C.1, we have $\|\nabla f(S_1^b, w, \kappa)\| \leq \kappa C_2 \sqrt{d}$ almost surely, and $\|\nabla f(S_2^b, w, \kappa)\| \leq \kappa C_2 \sqrt{d}$ almost surely. It follows that $\|Z^b\| \leq \kappa^2 C_2^2 d$. Therefore, $\|Z^b - \mathbb{E}[Z^b]\| \leq 2\kappa^2 C_2^2 d$.

Next, we will provide an upper bound for $\left\| \mathbb{E}\left[ \left(\sum_{b \in \widehat{B}}(Z^b - \mathbb{E}[Z^b])\right)\left(\sum_{b \in \widehat{B}}(Z^b - \mathbb{E}[Z^b])\right)^\intercal \right] \right\|$:

$$\left\| \mathbb{E}\left[ \left(\sum_{b \in \widehat{B}}(Z^b - \mathbb{E}[Z^b])\right)\left(\sum_{b \in \widehat{B}}(Z^b - \mathbb{E}[Z^b])\right)^\intercal \right] \right\|$$
$$= \left\| \mathbb{E}\left[ \sum_{b \in \widehat{B}}(Z^b - \mathbb{E}[Z^b])(Z^b - \mathbb{E}[Z^b])^\intercal \right] \right\|$$
$$\leq |\widehat{B}| \max_{b \in \widehat{B}} \left\| \mathbb{E}\left[ (Z^b - \mathbb{E}[Z^b])(Z^b - \mathbb{E}[Z^b])^\intercal \right] \right\|$$
$$\leq |\widehat{B}| \max_{b \in \widehat{B}} \left\| \mathbb{E}\left[ (Z^b(Z^b)^\intercal) \right] \right\|$$
$$\leq |\widehat{B}| \max_{b \in \widehat{B}} \left( \mathbb{E}[\|\nabla f(S_2^b, w, \kappa)\|^2] \cdot \left\| \mathbb{E}\left[ \nabla f(S_1^b, w, \kappa)\nabla f(S_1^b, w, \kappa)^\intercal \right] \right\| \right)$$
$$\leq |\widehat{B}| \max_{b \in \widehat{B}, u:\|u\|=1} \left( \mathbb{E}[\|\nabla f(S_2^b, w, \kappa)\|^2] \cdot \left\| \mathbb{E}\left[ (\nabla f(S_1^b, w, \kappa) \cdot u)^2 \right] \right\| \right).$$

From item 4 and item 5 in lemma C.1, we have:

$$\mathbb{E}[\|\nabla f(S_2^b, w, \kappa)\|^2] \leq C_1 \kappa^2 d,$$

and

$$\left\| \mathbb{E}\left[ (\nabla f(S_1^b, w, \kappa) \cdot u)^2 \right] \right\| \leq \kappa^2 C_1.$$

Combining these two bounds, wee obtain:

$$\left\| \mathbb{E}\left[ \left(\sum_{b \in \widehat{B}}(Z^b - \mathbb{E}[Z^b])\right)\left(\sum_{b \in \widehat{B}}(Z^b - \mathbb{E}[Z^b])\right)^\intercal \right] \right\| \leq |\widehat{B}| d\kappa^4 C_1^2.$$

Due to symmetry, the same bound holds for $\left\| \mathbb{E}\left[ \left(\sum_{b \in \widehat{B}}(Z^b - \mathbb{E}[Z^b])\right)^\intercal \left(\sum_{b \in \widehat{B}}(Z^b - \mathbb{E}[Z^b])\right) \right] \right\|$.

Finally, by applying the Matrix Bernstein inequality, we have:

$$\Pr\left[ \left\| \frac{1}{|\widehat{B}|} \sum_{b \in \widehat{B}}(Z^b - \mathbb{E}[Z^b]) \right\| \geq \alpha_s \epsilon^2 \kappa^2 C_1 \right] \leq 2d \exp\left\{ -\frac{|\widehat{B}|^2 \theta^2}{|\widehat{B}| d\kappa^4 C_1^2 + |\widehat{B}|\theta(2C_2^2 \kappa^2 d)} \right\}.$$

For $\widehat{B} = \Omega\left( \frac{d}{\alpha_s \epsilon^2}\left( \frac{1}{\alpha_s \epsilon^2} + \frac{C_2^2}{C_1} \right) \log \frac{d}{\delta'} \right)$, the quantity on the right-hand side is bounded by $\delta'$.

Therefore, with probability at least $1 - \delta'$, we have:

$$\left\| \frac{1}{|\widehat{B}|} \sum_{b \in \widehat{B}}(Z^b - \mathbb{E}[Z^b]) \right\| \leq \alpha_s \epsilon^2 \kappa^2 C_1.$$

Combining the above equation with Equation (23) completes the proof of the Theorem. $\square$

In the proof of Theorem E.1, we will utilize the following general linear algebraic result:

**Lemma E.3.** *For $z_0, z_1, ..., z_{k-1} \in \mathbb{R}^d$ and a probability distribution $(p_0, p_1, ..., p_{k-1})$ over $k$ elements, let $Z = \sum_{i=0}^{k-1} p_i z_i z_i^\mathsf{T}$. For a symmetric matrix $M$ and $\ell > 0$, let $u_1, u_2, .., u_\ell$ be top $\ell$ singular vectors of $M$ and let $U = [u_1, u_2, ..., u_\ell] \in \mathbb{R}^{d \times \ell}$, then we have:*

$$\|(I - UU^\mathsf{T})z_0\|^2 \leq \begin{cases} \frac{2(\ell+1)\|M-Z\|+\max_j \|z_j\|^2}{(\ell+1)p_0} & \ell < k \\ \frac{2\|M-Z\|}{p_0} & \text{if } \ell \geq k. \end{cases}$$

Lemma E.3 provides a bound on the preservation of the component $z_0$ by the subspace spanned by the top-$\ell$ singular vectors of a symmetric matrix $M$. This bound is expressed in terms of the spectral distance between matrices $Z$ and $M$, the maximum norm of any $z_i$, and the weight of the component corresponding to $z_0$ in $Z$. The proof of Lemma E.3 can be found in Section I.

Utilizing Lemma E.3 in conjunction with Theorem E.2, we proceed to prove Theorem E.1.

*Proof of Theorem E.1.* From Lemma E.3, we have the following inequality:

$$\|(I - UU^\mathsf{T})\mathbb{E}_{\mathcal{D}_0}[\nabla f(x,y,w,\kappa)]\|^2 \leq \begin{cases} \frac{2(\ell+1)\|A-\sum_{i=0}^{k-1} p_i z_i z_i^\mathsf{T}\|+\max_j \|\mathbb{E}_{\mathcal{D}_j}[\nabla f(x,y,w,\kappa)]\|^2}{(\ell+1)p_0} & \ell < k \\ \frac{2\|A-\sum_{i=0}^{k-1} p_i z_i z_i^\mathsf{T}\|}{p_0} & \text{if } \ell \geq k. \end{cases}$$

By applying Theorem E.2 and utilizing item 1 of Lemma C.1, it follows that with a probability of at least $1 - \delta'$, we have:

$$\|(I - UU^\mathsf{T})\mathbb{E}_{\mathcal{D}_0}[\nabla f(x,y,w,\kappa)]\|^2 \leq \begin{cases} \frac{2\alpha_s \epsilon^2 \kappa^2 C_1}{p_0} + \frac{\kappa^2 C_1}{(\ell+1)p_0} & \ell < k \\ \frac{2\alpha_s \epsilon^2 \kappa^2 C_1}{p_0} & \text{if } \ell \geq k. \end{cases}$$

The theorem then follows by using $p_0 \geq \alpha_s/2$ and $\ell \geq \min\{k, \frac{1}{2\alpha_s \epsilon^2}\}$. $\qquad\square$

# F  Grad Estimation

Recall that in gradient estimation for step $r$, Algorithm 1 utilizes the subroutine GRADSUBEST to find a projection matrix $P^{(r)}$ for an $\ell$-dimensional subspace. In the previous section, we showed that the difference between the expectation of the clipped gradient and the expectation of projection of the clipped gradient on the subspace for distribution $\mathcal{D}_0$ is small. Therefore, it suffices to estimate the expectation of projection of the clipped gradient on the subspace.

The main algorithm 1 passes the medium-sized batches $\widehat{B} = B_m^{(r)}$, the $\ell$-dimensional projection matrix $P = P^{(r)}$, and a collection of i.i.d. samples $S^* = S_2^{b^*,(r)}$ from $\mathcal{D}_0$ to the subroutine GRADEST. Here, $B_m^{(r)}$ is a random subset of the collection of medium-sized batches $B_m$.

The purpose of the GRADEST subroutine is to estimate the expected value of the projection of the clipped gradient onto the $\ell$-dimensional subspace defined by the projection matrix $P$. Since the subroutine operates on a smaller $\ell$-dimensional subspace, the minimum batch size required for the batches in $B_m$ and the number of batches required depend on $\ell$ rather than $d$.

The following theorem characterizes the final guarantee for the GRADEST subroutine:

**Theorem F.1.** *For subroutine GRADEST, let $n_m$ denote the length of the smallest batch in $\widehat{B}$, $N$ denote the number of batches in $\widehat{B}$ that has samples from $\mathcal{D}_0$ and $P$ be a projection matrix for some $\ell$ dimensional subspace of $\mathbb{R}^d$. If $T_1 \geq \Omega(\log \frac{|\widehat{B}|}{\delta'})$, $T_2 \geq \Omega(\log \frac{1}{\delta'})$, $n_m \geq 4T_1\Omega(\frac{\sqrt{\ell}}{\epsilon^2})$, $|S^*| \geq 2T_1\Omega(\frac{\sqrt{\ell}}{\epsilon^2})$ samples, and $N \cdot n_m \geq T_2\Omega(\frac{\ell}{\epsilon^2})$, then with probability $\geq 1 - 2\delta'$ the estimate $\Delta$ returned by subroutine GRADEST satisfy*

$$\|\Delta - \mathbb{E}_{\mathcal{D}_0}[P\nabla f(x,y,w,\kappa)]\| \leq 9\epsilon\kappa\sqrt{C_1}.$$

The above theorem implies that when the length of medium-sized batches is $\tilde{\Omega}(\sqrt{\ell})$ and the number of batches in $\widehat{B}$ containing samples from $\mathcal{D}_0$ is $\tilde{\Omega}(\ell)$, the GRADEST subroutine provides a reliable estimate of the projection of the clipped gradient onto the $\ell$-dimensional subspace defined by the projection matrix $P$.

Next, we provide a description of the GRADEST subroutine and present a brief outline of the proof for the theorem before formally proving it in the subsequent subsection.

In the GRADEST subroutine, the first step is to divide the samples in each batch of $\widehat{B}$ into two equal parts. By utilizing the first half of the samples in a batch $b$ along with the samples $S^*$, it estimates whether the expected values of the projection of the clipped gradient for $\mathcal{D}_0$ and the distribution used for the samples in $b$ are close or not. With high probability, the algorithm retains all the batches from $\mathcal{D}_0$ while rejecting batches from distributions where the difference between the two expectations is large. To achieve this with an $\ell$-dimensional subspace, we require $\tilde{\Omega}(\sqrt{\ell})$ samples in each batch (see Lemma F.6).

Following the rejection process, the GRADEST subroutine proceeds to estimate the projection of the clipped gradients within this $\ell$-dimensional subspace using the second half of the samples from the retained batches. To estimate the gradient accurately in the $\ell$-dimensional subspace, $\Omega(\ell)$ samples are sufficient (see Lemma F.7). To obtain guarantees with high probability, the procedure employs the median of means approach, both for determining which batches to keep and for estimation using the retained batches.

We prove the theorem formally in the next subsection.

## F.1 Proof of Theorem F.1

The following lemma provides an upper bound on the covariance of the projection of the clipped gradients.

**Lemma F.2.** *Consider a collection $S$ of $m$ i.i.d. samples from distribution $\mathcal{D}_i$. For $\kappa > 0$, $w \in \mathbb{R}^d$ and a projection matrix $P$ for an $\ell$ dimensional subspace of $\mathbb{R}^d$, we have*

$$\mathbb{E}[P\nabla f(S, w, \kappa)] = \mathbb{E}_{\mathcal{D}_i}[P\nabla f(x, y, w, \kappa)],$$

*and $\|Cov(P\nabla f(S, w, \kappa))\| \leq \frac{2\kappa^2}{m} C_1$ and $Tr\left(Cov(P\nabla f(S, w, \kappa))\right) \leq \ell \|Cov(P\nabla f(S, w, \kappa))\|$.*

*Proof.* Note that,

$$\mathbb{E}\left[P\left(\nabla f(S, w, \kappa)\right)\right] = P \,\mathbb{E}[\nabla f(S, w, \kappa)]] = P \,\mathbb{E}_{\mathcal{D}_i}[\nabla f(x, y, w, \kappa)] = \mathbb{E}_{\mathcal{D}_i}[P\nabla f(x, y, w, \kappa)],$$

where the second-to-last equality follows from Lemma C.1.

This proves the first part of the lemma. To prove the second part, we bound the norm of the covariance matrix:

$$
\begin{aligned}
\mathrm{Cov}(P\nabla f(S, w, \kappa)) &= \max_{\|u\| \leq 1} \mathrm{Var}(u^\mathsf{T} P \nabla f(S, w, \kappa)) \\
&= \max_{\|v\| \leq 1} \mathrm{Var}(v^\mathsf{T} \nabla f(S, w, \kappa)) \\
&\leq \|\mathrm{Cov}(\nabla f(S, w, \kappa))\| \\
&\leq \frac{\kappa^2}{m} C_1,
\end{aligned}
$$

where the last inequality follows from Lemma C.1. Similarly,

$$\mathrm{Cov}(P\nabla f(S', w, \kappa)) \leq \frac{\kappa^2}{m} C_1.$$

Finally, since random vector $P\nabla f(S', w, \kappa)$ lies in $\ell$ dimensional subspace of $\mathbb{R}^d$, corresponding to projection matrix $P$, hence its covariance matrix has rank $\leq \ell$. Hence, the relation $\mathrm{Tr}\left(\mathrm{Cov}(P\nabla f(S, w, \kappa))\right) \leq \ell \|\mathrm{Cov}(P\nabla f(S, w, \kappa))\|$ follows immediately. $\square$

The following corollary is a simple consequence of the previous lemma:

**Corollary F.3.** *Consider two collections $S$ and $S'$ each consisting of $m$ i.i.d. samples from distributions $\mathcal{D}_i$ and $\mathcal{D}_0$, respectively. For $\kappa > 0$, $w \in \mathbb{R}^d$ and a projection matrix $P$ for an $\ell$ dimensional subspace of $\mathbb{R}^d$, let $z = P\left(\nabla f(S, w, \kappa) - \nabla f(S', w, \kappa)\right)$, we have:*

$$\mathbb{E}[z] = \mathbb{E}_{\mathcal{D}_i}[P\nabla f(x, y, w, \kappa)] - \mathbb{E}_{\mathcal{D}_0}[P\nabla f(x, y, w, \kappa)],$$

*and $\|Cov(z)\| \leq \frac{4\kappa^2}{m} C_1$ and $Tr\left(Cov(z)\right) \leq \ell \|Cov(z)\|$.*

*Proof.* The expression for $\mathbb{E}[z]$ can be obtained from the previous lemma and the linearity of expectation.

To prove the second part, we bound the norm of the covariance matrix of $z$.

$$\mathrm{Cov}(z) = \mathrm{Cov}(P(\nabla f(S, w, \kappa) - \nabla f(S', w, \kappa))) \leq 2(\mathrm{Cov}(P\nabla f(S, w, \kappa)) + \mathrm{Cov}(P\nabla f(S', w, \kappa))).$$

Using the bounds from the previous lemma, we can conclude that $\|\mathrm{Cov}(z)\| \leq \frac{4\kappa^2}{m}C_1$.

Finally, since the random vector $z$ lies in the $\ell$-dimensional subspace of $\mathbb{R}^d$ defined by the projection matrix $P$, its covariance matrix has rank $\leq \ell$. Therefore, we have $\mathrm{Tr}\,(\mathrm{Cov}(z)) \leq \ell\|\mathrm{Cov}(z)\|$. $\qquad\square$

The following theorem bounds the variance of the dot product of two independent random vectors. It will be helpful in upper bounding the variance of $\zeta_j^b$ (defined in subroutine GRADEST).

**Theorem F.4.** *For any two independent random vectors $z_1$ and $z_2$, we have:*

$$\mathrm{Var}(z_1 \cdot z_2) \leq 3Tr(Cov(z_1)) \cdot \|Cov(z_2)\| + 3\|\mathbb{E}[z_1]\|^2\|Cov(z_2)\| + 3\|\mathbb{E}[z_2]\|^2\|Cov(z_1)\|.$$

*Proof.* We start by expanding the variance expression:

$$\begin{aligned}
\mathrm{Var}(z_1 \cdot z_2) &= \mathrm{Var}(z_1 \cdot z_2 - \mathbb{E}[z_1 \cdot z_2]) \\
&= \mathrm{Var}(z_1 \cdot z_2 - \mathbb{E}[z_1] \cdot \mathbb{E}[z_2]) \\
&= \mathrm{Var}((z_1 - \mathbb{E}[z_1]) \cdot (z_2 - \mathbb{E}[z_2]) + \mathbb{E}[z_1] \cdot z_2 + \mathbb{E}[z_2] \cdot z_1) \\
&\leq 3\,\mathrm{Var}((z_1 - \mathbb{E}[z_1]) \cdot (z_2 - \mathbb{E}[z_2])) + 3\,\mathrm{Var}(\mathbb{E}[z_1] \cdot z_2) + 3\,\mathrm{Var}(\mathbb{E}[z_2] \cdot z_1) \\
&= 3\,\mathrm{Var}((z_1 - \mathbb{E}[z_1]) \cdot (z_2 - \mathbb{E}[z_2])) + 3\,\mathbb{E}[z_1]^\mathsf{T}\mathrm{Cov}(z_2)\,\mathbb{E}[z_1] + \mathbb{E}[z_2]^\mathsf{T}\mathrm{Cov}(z_1)\,\mathbb{E}[z_2] \\
&\leq 3\,\mathrm{Var}((z_1 - \mathbb{E}[z_1]) \cdot (z_2 - \mathbb{E}[z_2])) + 3\|\mathbb{E}[z_1]\|^2\|\mathrm{Cov}(z_2)\| + 3\|\mathbb{E}[z_2]\|^2\|\mathrm{Cov}(z_1)\|.
\end{aligned}$$

To complete the proof, we bound the first term in the last expression:

$$\begin{aligned}
\mathrm{Var}((z_1 - \mathbb{E}[z_1]) \cdot (z_2 - \mathbb{E}[z_2])) &= \mathbb{E}\big[((z_1 - \mathbb{E}[z_1]) \cdot (z_2 - \mathbb{E}[z_2]))^2\big] \\
&= \mathbb{E}[(z_1 - \mathbb{E}[z_1])^\mathsf{T}\mathrm{Cov}(z_2)(z_1 - \mathbb{E}[z_1])] \\
&\leq \mathbb{E}\big[\|z_1 - \mathbb{E}[z_1]\|^2\big] \cdot \|\mathrm{Cov}(z_2)\| \\
&= \mathrm{Tr}(\mathrm{Cov}(z_1)) \cdot \|\mathrm{Cov}(z_2)\|.
\end{aligned}$$

$\qquad\square$

Using the two previous results, we can establish a bound on the expectation and variance of $\zeta_j^b$.

**Lemma F.5.** *In subroutine GRADEST, let $P$ be a projection matrix of an $\ell$ dimensional subspace. Suppose $S_j^*$ has $\geq m$ i.i.d. samples from $\mathcal{D}_0$ and, $S_{1,j}^b$ and $S_{1,j+T_1}^b$ have $\geq m$ i.i.d. samples from $\mathcal{D}_i$ for some $i \in \{0, 1, ..., k-1\}$. Than we have:*

$$\mathbb{E}[\zeta_j^b] = \left\|\mathbb{E}_{\mathcal{D}_i}[P\nabla f(x, y, w, \kappa)] - \mathbb{E}_{\mathcal{D}_0}[P\nabla f(x, y, w, \kappa)]\right\|^2$$

*and*

$$\mathrm{Var}(\zeta_j^b) \leq \frac{48}{m^2}\kappa^4\ell C_1{}^2 + \frac{24}{m}\kappa^2\,\mathbb{E}[\zeta_j^b]C_1.$$

*Proof.* Let $z_1 = P(\nabla f(S_{1,j}^b, w, \kappa) - \nabla f(S_j^*, w, \kappa))$ and $z_2 = P(\nabla f(S_{1,T_1+j}^b, w, \kappa) - \nabla f(S_{T_1+j}^*, w, \kappa))$.

From Corollary F.3, we know that

$$\mathbb{E}[z_1] = \mathbb{E}[z_2] = \mathbb{E}_{\mathcal{D}_i}[P\nabla f(x, y, w, \kappa)] - \mathbb{E}_{\mathcal{D}_0}[P\nabla f(x, y, w, \kappa)],$$

$$\mathrm{Cov}(z_1) = \mathrm{Cov}(z_2) = \frac{4\kappa^2}{m}C_1$$

and

$$\mathrm{Tr}(\mathrm{Cov}(z_1)) = \mathrm{Tr}(\mathrm{Cov}(z_2)) = \frac{4}{m}\ell\kappa^2 C_1.$$

Note that $\zeta_j^b = z_1 \cdot z_2$. Then bound on the variance of $\zeta_j^b$ follows by combining the above bounds with Theorem F.4. Finally, the expected value of $\zeta_j^b$ is:

$$\mathbb{E}[\zeta_j^b] = \mathbb{E}[z_1] \cdot \mathbb{E}[z_2] = \|\mathbb{E}[z_1]\|^2.$$

$\square$

The following lemma provides a characterization of the minimum batch length in $\widehat{B}$ and the size of the collection $S^*$ required for successful testing in subroutine GRADEST.

**Lemma F.6.** *In subroutine* GRADEST*, let $P$ be a projection matrix of an $\ell$ dimensional subspace, $T_1 \geq \Omega(\log \frac{|\widehat{B}|}{\delta'})$, and each batch $b \in \widehat{B}$ has at least $|S^b| \geq 4T_1\Omega(\frac{\sqrt{\ell}}{\epsilon^2})$ samples, and $|S^*| = 2T_1\Omega(\frac{\sqrt{\ell}}{\epsilon^2})$. Then with probability $\geq 1 - \delta'$, the subset $\tilde{B}$ in subroutine* GRADEST *satisfy the following:*

1. $|\tilde{B}|$ *retains all the batches in $\widehat{B}$ that had samples from $\mathcal{D}_0$.*

2. $\tilde{B}$ *does not contain any batch that had samples from $\mathcal{D}_i$ if $i$ is such that*

$$\|\mathbb{E}_{\mathcal{D}_i}[P\nabla f(x, y, w, \kappa)] - \mathbb{E}_{\mathcal{D}_0}[P\nabla f(x, y, w, \kappa)]\| > 2\epsilon\kappa\sqrt{C_1}.$$

*Proof.* The lower bound on $|S^b|$ in the lemma ensures that for each batch $b$ and all $j \in [2T_1]$, we have $S_{1,j}^b = \Omega(\frac{\sqrt{\ell}}{\epsilon^2})$, and the lower bound on $|S^|$ ensures that for all $j \in [2T_1]$, $S_j = \Omega(\frac{\sqrt{\ell}}{\epsilon^2})$.

First, consider the batches that have samples from the distribution $\mathcal{D}_0$.

For any such batch $b$ and $j \in [T_1]$, from Lemma F.5, we have $\mathbb{E}[\zeta_j^b] = 0$ and $\mathrm{Var}(\zeta_j^b) = \mathcal{O}(\epsilon^4\kappa^2{C_1}^2)$. Therefore, for $T_1 \geq \Omega(\log \frac{|\widehat{B}|}{\delta'})$, it follows that with probability $\geq 1 - \delta'/2$ for every batch $b \in \widehat{B}$ that has samples from $\mathcal{D}_0$ the median of $\{\zeta_j^b\}_{j \in [T_1]}$ will be less than $\epsilon^2\kappa^2 C_1$, and it will be retained in $\tilde{B}$. This completes the proof of the first part.

Next, consider the batches that have samples from any distribution $\mathcal{D}_i$ for which

$$\|\mathbb{E}_{\mathcal{D}_i}[P\nabla f(x, y, w, \kappa)] - \mathbb{E}_{\mathcal{D}_0}[P\nabla f(x, y, w, \kappa)]\| > 2\epsilon\kappa\sqrt{C_1}.$$

For any such batch $b$ and $j \in [T_1]$, according to Lemma F.5, we have $\mathbb{E}[\zeta_j^b] \geq 4\epsilon^2\kappa^2 C_1$ and $\mathrm{Var}(\zeta_j^b) = \mathcal{O}(\mathbb{E}[\zeta_j^b]^2)$. Hence, for $T_1 \geq \Omega(\log \frac{|\widehat{B}|}{\delta'})$, it follows that with probability at least $1 - \delta'/2$, the median of $\{\zeta_j^b\}_{j \in [T_1]}$ for every batch will be greater than $\epsilon^2\kappa^2 C_1$, and those batches will not be included in $\tilde{B}$. This completes the proof of the second part. $\square$

The following theorem characterizes the number of samples required in $\tilde{B}$ for an accurate estimation of $\Delta$.

**Lemma F.7.** *Suppose the conclusions in Lemma F.6 hold for $\tilde{B}$ defined in subroutine* GRADEST*, $T_2 \geq \Omega(\log \frac{1}{\delta'})$, each batch $b \in \tilde{B}$ has size $\geq n_m$, and $|\tilde{B}| \cdot n_m \geq 2T_2\Omega(\frac{\ell}{\epsilon^2})$, then with probability $\geq 1 - \delta'$ the estimate $\Delta$ returned by subroutine* GRADEST *satisfy*

$$\|\Delta - \mathbb{E}_{\mathcal{D}_0}[P\nabla f(x, y, w, \kappa)]\| \leq 9\epsilon\kappa\sqrt{C_1}.$$

*Proof.* Recall that in subroutine GRADEST, we defined

$$\Delta_i = \frac{1}{|\tilde{B}|}\sum_{b \in \tilde{B}} P\nabla f(S_{2,i}^b, w, \kappa).$$

Let $z_i^b = P\nabla f(S_{2,i}^b, w, \kappa)$. From Lemma F.6, for all $b \in \tilde{B}$, we have

$$\left\|\mathbb{E}[z_i^b] - \mathbb{E}_{\mathcal{D}_0}[P\nabla f(x, y, w, \kappa)]\right\| \leq 2\epsilon\kappa\sqrt{C_1}.$$

Therefore,

$$\| \mathbb{E}[\Delta_i] - \mathbb{E}_{\mathcal{D}_0}[P\nabla f(x,y,w,\kappa)]\| = \left\| \frac{1}{|\tilde{B}|} \sum_{b \in \tilde{B}} \mathbb{E}[z_i^b] - \mathbb{E}_{\mathcal{D}_0}[P\nabla f(x,y,w,\kappa)] \right\|$$

$$\leq \max_{b \in \tilde{B}} \| \mathbb{E}[z_i^b] - \mathbb{E}_{\mathcal{D}_0}[P\nabla f(x,y,w,\kappa)]\| \leq 2\epsilon\kappa\sqrt{C_1}. \quad (24)$$

Next, from Lemma F.2,

$$\|\mathrm{Cov}(z_i^b)\| \leq \frac{\kappa^2}{|S_{2,i}^b|}C_1 = \frac{T_2\kappa^2}{|S_2^b|}C_1 = \frac{2T_2\kappa^2}{|S^b|}C_1 \leq \frac{2T_2\kappa^2 C_1}{\min_{b \in \tilde{B}} |S^b|}, \quad (25)$$

where the two equalities follow because for all batches $b \in \tilde{B}$, $|S_{2,i}^b| = |S_2^b|/T_2$ and $|S_2^b| = |S^b|/2$.
Then

$$\|\mathrm{Cov}(\Delta_i)\| = \frac{1}{|\tilde{B}|} \max_{b \in \tilde{B}} \|\mathrm{Cov}(z_i^b)\| \leq \frac{2T_2\kappa^2 C_1}{|\tilde{B}| \cdot \min_{b \in \tilde{B}} |S^b|}$$

Since $\Delta_i$ lies in an $\ell$ dimensional subspace of $\mathbb{R}^d$, it follows that

$$\mathrm{Tr}(\mathrm{Cov}(\Delta_i)) \leq \ell\|\mathrm{Cov}(\Delta_i)\| \leq \frac{2\ell T_2\kappa^2 C_1}{|\tilde{B}| \cdot \min_{b \in \tilde{B}} |S^b|}$$

Note that $\mathrm{Var}(\|\Delta_i - \mathbb{E}[\Delta_i]\|) = \mathrm{Tr}(\mathrm{Cov}(\Delta_i))$. Then, from Chebyshev's bound:

$$\Pr[\|\Delta_i - \mathbb{E}[\Delta_i]\| \geq \epsilon\kappa\sqrt{C_1}] \leq \frac{\mathrm{Var}(\|\Delta_i - \mathbb{E}[\Delta_i]\|)}{\epsilon^2\kappa^2 C_1} \leq \frac{2\ell T_2}{\epsilon^2|\tilde{B}| \cdot \min_{b \in \tilde{B}} |S^b|} \leq 1/8.$$

Combining above with Equation (24),

$$\Pr[\|\Delta_i - \mathbb{E}_{\mathcal{D}_0}[P\nabla f(x,y,w,\kappa)]\| \geq 3\epsilon\kappa\sqrt{C_1}] \leq 1/4.$$

Let $D := \{i \in [T_2] : \|\Delta_i - \mathbb{E}_{\mathcal{D}_0}[P\nabla f(x,y,w,\kappa)]\| \leq 3\epsilon\kappa\sqrt{C_1}\|\}$. Then, for $T_2 = \Omega(\log \frac{1}{\delta'})$, with probability $\geq 1 - \delta'$, we have

$$|D| \geq \frac{1}{2}T_2.$$

Recall that in the subroutine, we defined $\xi_i = median\{j \in [T_2] : \|\Delta_i - \Delta_j\|\}$ and $i^* = \arg\min\{i \in [T_2] : \xi_i\}$.

From the definition of $D$, and triangle inequality, for all $i, j \in D$, we have $\|\Delta_i - \Delta_j\| \leq 6\epsilon\kappa\sqrt{C_1}$. Therefore, if $|D| \geq \frac{1}{2}T_2$, then for any $i \in D$, $\xi_i \leq 6\epsilon\kappa\sqrt{C_1}$. This would imply $\xi_{i^*} \leq 6\epsilon\kappa\sqrt{C_1}$. Furthermore, since $|D| \geq \frac{1}{2}T_2$, there exist at least one $i \in D$ such that $\|\Delta_i - \Delta_{i^*}\| \leq 6\epsilon\kappa\sqrt{C_1}$. Using the definition of $D$, and the triangle inequality, we can conclude that

$$\|\Delta i^* - \mathbb{E}_{\mathcal{D}_0}[P\nabla f(x,y,w,\kappa)]\| \leq \|\Delta_i - \mathbb{E}_{\mathcal{D}_0}[P\nabla f(x,y,w,\kappa)]\| + \|\Delta_i - \Delta_{i^*}\| \leq 9\epsilon\kappa\sqrt{C_1}.$$

$\square$

Theorem F.1 then follows by combining lemmas F.6 and F.7.

# G  Number of steps required

The following lemma shows that with a sufficiently accurate estimation of the expectation of gradients, a logarithmic number of gradient descent steps are sufficient in the main algorithm 1.

**Lemma G.1.** *For $\epsilon > 0$, suppose $\|\Delta^{(r)} - \Sigma_0(\hat{w}^{(r)} - w_0)\| \leq \frac{1}{2}\|\hat{w}^{(r)} - w_0\| + \frac{\epsilon\sigma}{4}$, and $R = \Omega(C_1 \log \frac{\|w_0\|}{\sigma})$, then $\|\hat{w}^{(r)} - w_0\| \leq \epsilon\sigma$.*

*Proof.* Recall that $\hat{w}^{(r+1)} = \hat{w}^{(r)} - \frac{1}{C_1}\Delta^{(r)}$. Then we have:

$$\hat{w}^{(r+1)} - w_0 = \hat{w}^{(r)} - w_0 - \frac{1}{C_1}\Delta^{(r)} = (\hat{w}^{(r)} - w_0)\left(I - \frac{1}{C_1}\Sigma_0\right) + \frac{1}{C_1}(\Sigma_0(\hat{w}^{(R+1)} - w_0) - \Delta^{(r)}).$$

Using triangle inequality, we obtain:

$$\begin{aligned}
\|\hat{w}^{(r+1)} - w_0\| &\leq \|\hat{w}^{(r)} - w_0\|\left\|I - \frac{1}{C_1}\Sigma_0\right\| + \frac{1}{C_1}\|\Sigma_0(\hat{w}^{(r)} - w_0) - \Delta^{(r)}\| \\
&\leq \|\hat{w}^{(r)} - w_0\|\left(1 - \frac{1}{C_1}\right) + \frac{1}{C_1}\left(\frac{\|\hat{w}^{(r)} - w_0\|}{2} + \frac{\epsilon\sigma}{4}\right) \\
&\leq \|\hat{w}^{(r)} - w_0\|\left(1 - \frac{1}{2C_1}\right) + \frac{\epsilon\sigma}{4C_1}.
\end{aligned}$$

Using recursion, we have:

$$\begin{aligned}
\|\hat{w}^{(R+1)} - w_0\| &\leq \|\hat{w}^{(1)} - w_0\|\left(1 - \frac{1}{2C_1}\right)^R + \sum_{i=0}^{R-1}\left(1 - \frac{1}{2C_1}\right)^i \frac{\epsilon\sigma}{4C_1} \\
&\leq \|\hat{w}^{(1)} - w_0\|\exp\left(-\frac{R}{2C_1}\right) + 2C_1\frac{\epsilon\sigma}{4C_1} \\
&\leq \epsilon\sigma,
\end{aligned}$$

where the second inequality follows from the upper bound on the sum of infinite geometric series and the last inequality follows from the bound on $R$ and $\hat{w}^{(1)} = 0$. $\qquad\square$

## H   Final Estimation Guarantees

*Proof of Theorem 2.1.* We show that with probability $\geq 1 - \delta$, for each $r \in [R]$, the gradient computed by the algorithm satisfies $\|\Delta^{(r)} - \Sigma_0(\hat{w}^{(r)} - w_0)\| \leq \frac{1}{2}\|\hat{w}^{(r)} - w_0\| + \frac{\epsilon\sigma}{4}$. Lemma G.1 then implies that for $R = \Omega(C_1 \log \frac{M}{\sigma})$, the output returned by the algorithm $\hat{w} = \hat{w}^{(R+1)}$ satisfy $\|\hat{w} - w_0\| \leq \epsilon\sigma$.

To show this, we fix $r$, and for this value of $r$, we show that with probability $\geq 1 - \delta/R$, $\|\Delta^{(r)} - \Sigma_0(\hat{w}^{(r)} - w_0)\| \leq \frac{1}{2}\|\hat{w}^{(r)} - w_0\| + \frac{\epsilon\sigma}{4}$. Since each round uses an independent set of samples, the theorem then follows by applying the union bound.

First, we determine the bound on the clipping parameter. From Theorem D.1, for $|S^{b^*}|/R = \Omega(C^2 \log 1/\delta')$, with probability $\geq 1 - \delta'$, we have

$$\sqrt{\frac{8(C+1)C_1\left(\mathbb{E}_{\mathcal{D}_0}[(y - x \cdot w^{(r)})^2]\right)}{\epsilon_1}} \leq \kappa^{(r)} \leq 28\sqrt{\frac{2(C+1)C_1\left(\mathbb{E}_{\mathcal{D}_0}[(x \cdot (w^{(r)} - w_0))^2] + \sigma^2\right)}{\epsilon_1}}. \tag{26}$$

Next, employing Theorem C.2 and utilizing the lower bound on the clipping parameter in the above equation, we obtain the following bound on the norm of the expected difference between clipped and unclipped gradients:

$$\left\|\mathbb{E}_{\mathcal{D}_0}[(\nabla f(x, y, w^{(r)}, \kappa^{(r)}) - \Sigma_0(w^{(r)} - w_0)\right\| \leq \epsilon_1\|w^{(r)} - w_0\|. \tag{27}$$

Recall that in $B_s$, at least $\alpha_s$ fraction of the batches contain samples from $\mathcal{D}_0$. When $B_s$ is divided into $R$ equal random parts, w.h.p. each part $B_s^{(r)}$ will have at least $\alpha_s$ fraction of the batches containing samples from $\mathcal{D}_0$.

From Theorem E.1, if $|B_s^{(r)}| = \frac{|B_s|}{R} = \Omega\left(\frac{d}{\alpha_s\epsilon_2^2}\left(\frac{1}{\alpha_s\epsilon_2^2} + \frac{C_2^2}{C_1}\right)\log\frac{d}{\delta'}\right)$, then with probability $\geq 1 - \delta'$, the projection matrix $P^{(r)}$ satisfies

$$\|\mathbb{E}_{\mathcal{D}_0}[\nabla f(x, y, w^{(r)}, \kappa^{(r)})] - P^{(r)}\mathbb{E}_{\mathcal{D}_0}[\nabla f(x, y, w^{(r)}, \kappa^{(r)})]\| \leq 4\epsilon_2\kappa^{(r)}\sqrt{C_1}. \tag{28}$$

The above equation shows subroutine GRADSUBEST finds projection matrix $P^{(r)}$ such that the expected value of clipped gradients projection is roughly the same as the expected value of the clipped gradient.

Next, we show that subroutine GRADEST provides a good estimate of the expected value of clipped gradients projection. Let $N$ denote the number of batches in $B_m$ that have samples from $\mathcal{D}_0$. If $N \geq \Omega(R + \log 1/\delta')$ then with probability $\geq 1 - \delta'$, $B_m^{(r)}$ has $\Theta(N/R)$ batches sampled from $\mathcal{D}_0$. If each batch in $B_m$ and batch $b^*$ has more than $n_m$ samples, $\frac{n_m}{R} = \Omega(\frac{\sqrt{\ell}}{\epsilon_2^2}\log(\frac{|B_m|}{\delta'}))$, and $\frac{N \cdot n_m}{R} \geq \Omega(\frac{\ell}{\epsilon_2^2}\log 1/\delta')$, then from Theorem F.1, with probability $\geq 1 - \delta'$

$$\left\| \Delta^{(r)} - \mathbb{E}_{\mathcal{D}_0}[P^{(r)}\nabla f(x, y, w^{(r)}, \kappa^{(r)})] \right\| \leq 9\epsilon_2\kappa^{(r)}\sqrt{C_1}. \tag{29}$$

Combining the above three equations using triangle inequality,

$$\|\Delta^{(r)} - \Sigma_0(\hat{w}^{(r)} - w_0)\| \leq 13\epsilon_2\kappa^{(r)}\sqrt{C_1} + \epsilon_1\|w^{(r)} - w_0\|, \tag{30}$$

with probability $\geq 1 - 5\delta'$.

In equation (26) using the upper bound, $\mathbb{E}_{\mathcal{D}_0}[(x \cdot (w^{(r)} - w_0))^2] \leq \|w^{(r)} - w_0\|^2\|\Sigma_0\| \leq C_1\|w^{(r)} - w_0\|^2$ we get

$$\kappa^{(r)} \leq 28\sqrt{\frac{2(C+1)C_1^2\|w^{(r)} - w_0\|^2 + (C+1)C_1\sigma^2}{\sqrt{\epsilon_1}}} \leq \frac{28\sqrt{2(C+1)}}{\sqrt{\epsilon_1}}(C_1\|w^{(r)} - w_0\| + \sqrt{C_1}\sigma).$$

Combining the two equations,

$$\|\Delta^{(r)} - \Sigma_0(\hat{w}^{(r)} - w_0)\| \leq \frac{364\epsilon_2\sqrt{2(C+1)C_1}}{\sqrt{\epsilon_1}}(C_1\|w^{(r)} - w_0\| + \sqrt{C_1}\sigma) + \epsilon_1\|w^{(r)} - w_0\|, \tag{31}$$

with probability $\geq 1 - 5\delta'$ There exist universal constants $c_1, c_2 > 0$ such that for $\epsilon_1 = c_1$ and $\epsilon_2 = \frac{c_2}{C_1\sqrt{C+1}}\left(\epsilon + \frac{1}{\sqrt{C_1}}\right)$, the quantity on the right is bounded by $\|w^{(r)} - w_0\|/2 + \epsilon\sigma/4$. We choose these values for $\epsilon_1$ and $\epsilon_2$ and $\delta' = \frac{\delta}{5R}$.

From the above discussion, it follows that if $|B_s| = \tilde{\Omega}\left(\frac{d}{\alpha_s^2\epsilon^4}\right)$, $n_m \geq \tilde{\Omega}(\frac{\sqrt{\ell}}{\epsilon^2})$, and $B_m$ has $\geq \frac{1}{n_m}\tilde{\Omega}\left(\frac{\ell}{\epsilon^2}\right)$ batches sampled from $\mathcal{D}_0$, then with probability $\geq 1 - \delta/R$,

$$\|\Delta^{(r)} - \Sigma_0(\hat{w}^{(r)} - w_0)\| \leq \frac{1}{2}\|\hat{w}^{(r)} - w_0\| + \frac{\epsilon\sigma}{4}.$$

Using $\ell = \min\{k, \frac{1}{\epsilon^2\alpha_s}\}$, we get the bounds on the number of samples and batches required by the algorithm. $\qquad\square$

## I  Proof of Lemma E.3

To establish the lemma, we first introduce and prove two auxiliary lemmas.

**Lemma I.1.** *For $k > 0$, and a probability distribution $(p_0, p_1, ..., p_{k-1})$ over $k$ elements, let $Z = \sum_{i=0}^{k-1} p_0 z_i z_i^\mathsf{T}$, where $z_i$ are $d$-dimensional vectors. Then for all $\ell \geq 0$, $\ell^{th}$ largest singular value of $Z$ is bounded by $\max_i \|z_i\|^2/\ell$.*

*Proof.* Note that $Z$ is a symmetric matrix, so its left and right singular values are the same. Let $v_1, v_2, ...$ be the singular vectors in the SVD decomposition of $Z$, and let $a_1 \leq a_2 \leq a_3 \leq ...$ be the corresponding singular values. Using the properties of SVD, we have:

$$\sum_i a_i = \sum_i v_i^\mathsf{T} Z v_i = \sum_i v_i^\mathsf{T}\left(\sum_{j=0}^{k-1} p_j z_j z_j^\mathsf{T}\right)v_i = \sum_{j=0}^{k-1} p_j \sum_i (v_i \cdot z_j)^2 \leq \sum_{j=0}^{k-1} p_j\|z_j\|^2 \leq \max_j\|z_j\|^2.$$

Next, we have:

$$\sum_i a_i \geq \sum_{i \leq \ell} a_i \geq \sum_{i \leq \ell} a_\ell = \ell \cdot a_\ell.$$

Combining the last two equations yields the desired result. □

**Lemma I.2.** *Let* $u_1, u_2, .., u_\ell \in \mathbb{R}^d$ *be* $\ell$ *mutually orthogonal unit vectors, and let* $U = [u_1, u_2, ..., u_\ell] \in \mathbb{R}^{d \times \ell}$. *For any set of* $k$ *vectors* $z_0, z_1, ..., z_{k-1} \in \mathbb{R}^d$, *non-negative reals* $p_0, p_1, ..., p_{k-1}$, *and reals* $a_1, a_2, ..., a_\ell$, *we have:*

$$\|(I - UU^\intercal)z_0\|^2 \leq \frac{\left\|\sum_{i=1}^{k-1} p_i z_i z_i^\intercal - \sum_{j \in [\ell]} a_j u_j u_j^\intercal\right\|}{p_0}.$$

*Proof.* Let $v = (I - UU^\intercal)z_0$. First we show that for all $j \in [\ell]$, the vectors $v$ and $u_j$ are orthogonal,

$$u_j^\intercal(I - UU^\intercal)z_0 = (u_j^\intercal \cdot z_0) - (u_j^\intercal \cdot z_0) = 0.$$

Then,

$$\left\|v^\intercal\left(\sum_{i=0}^{k-1} p_i z_i z_i^\intercal - \sum_{j \in [\ell]} a_j u_j u_j^\intercal\right)v\right\| = \left\|v^\intercal\left(\sum_{i=0}^{k-1} p_i z_i z_i^\intercal\right)v\right\| = \left\|\sum_{i=0}^{k-1} p_i(z_i^\intercal v)^2\right\| \geq \left\|p_0(z_0^\intercal v)^2\right\|$$

Next, we have:

$$z_0^\intercal v = z_0^\intercal(I - UU^\intercal)v + z_0 UU^\intercal v = z_0^\intercal(I - UU^\intercal)v = v^\intercal v = \|v\|^2$$

Combining the last two equations, we obtain:

$$\|v\|^2 \cdot \left\|\sum_{i=0}^{k-1} p_i z_i z_i^\intercal - \sum_{j \in [\ell]} a_j u_j u_j^\intercal\right\| \geq \left\|v^\intercal\left(\sum_{i=0}^{k-1} z_i z_i^\intercal - \sum_{j \in [\ell]} a_j u_j u_j^\intercal\right)v\right\| \geq p_0\|v\|^4.$$

Dividing both sides by $\|v\|^2$ completes the proof. □

Next, combining the above two auxiliary lemmas we prove Lemma E.3.

*Proof of Lemma E.3.* Let $\Lambda_i(\cdot)$ denote the $i^{th}$ largest singular value of a matrix. Let $\hat{M}$ be rank $\ell$ truncated-SVD of $M$, then it follows that,

$$\|M - \hat{M}\| = \Lambda_{\ell+1}(M).$$

First, we consider the case $\ell < k$. By applying Weyl's inequality for singular values, we have

$$\Lambda_{\ell+1}(M) \leq \Lambda_{\ell+1}(Z) + \Lambda_1(M - Z) \leq \frac{\max_j \|z_j\|^2}{\ell + 1} + \|M - Z\|,$$

where the last equation follows from Lemma I.1.

First applying the triangle inequality, and then using the above two equations, we have

$$\|\hat{M} - Z\| \leq \|M - \hat{M}\| + \|M - Z\| \leq \frac{\max_j \|z_j\|^2}{\ell + 1} + 2\|M - Z\|.$$

Combining the above equation with Lemma I.2, we have:

$$\|(I - UU^\intercal)z_0\|^2 \leq \frac{2(\ell + 1)\|M - Z\| + \max_j \|z_j\|^2}{(\ell + 1)p_0}.$$

This completes the proof for $\ell < k$. To prove for the case $\ell > k$, we use $\Lambda_{\ell+1}(Z) = 0$ in place of the bound $\Lambda_{\ell+1}(Z) \leq \frac{\max_j \|z_j\|^2}{\ell+1}$ in the above proof for the case $\ell < k$. □

## J  Removing the Additional Assumptions

To simplify our analysis, we made two assumptions about the data distributions. We now argue that these assumptions are not limiting.

The first additional assumption was that there exists a constant $C_2 > 0$ such that for all components $i \in \{0, 1, \ldots, k-1\}$ and random samples $(x, y) \sim \mathcal{D}_i$, we have $\|x\| \leq C_2 \sqrt{d}$ almost surely. In the non-batch setting, Cherapanamjeri et al. (2020) [CAT$^+$20] have shown that this assumption is not limiting. They showed that if other assumptions are satisfied, then there exists a constant $C_2$ such that with probability $\geq 0.99$, we have $\|x\| \leq C_2 \sqrt{d}$. Therefore, disregarding the samples for which $|x| > C_2 \sqrt{d}$ does not significantly reduce the data size. Moreover, it has minimal impact on the covariance matrix and hypercontractivity constants of the distributions. This argument easily extends to the batch setting. In the batch setting, we first exclude samples from batches where $\|x\| > C_2 \sqrt{d}$. Then we remove small-sized batches with fewer than or equal to 2 samples and medium-sized batches that have been reduced by more than 10% of their original size. It is easy to show that w.h.p. the fraction of medium and small size batches that gets removed for any component is at most 10%. THence, this assumption can be removed with a small increase in the batch size and the number of required samples in our main results.

Next, we address the assumption that the noise distribution is symmetric. We can handle this by employing a simple trick. Consider two independent and identically distributed (i.i.d.) samples $(x_1, y_1)$ and $(x_2, y_2)$, where $y_i = w^* \cdot x_i + \eta_i$. We define $x = (x_1 - x_2)/\sqrt{2}$, $y = (y_1 - y_2)/\sqrt{2}$, and $\eta = (\eta_1 - \eta_2)/\sqrt{2}$. It is important to note that the distribution of $\eta$ is symmetric around 0, and the covariance of $x$ is the same as that of $x_i$, while the variance of $\eta$ is the same as that of $\eta_i$. Furthermore, we have $y = w^* \cdot x + \eta$. Therefore, the new sample $(x, y)$ obtained by combining two i.i.d. samples satisfies the same distributional assumptions as before, and in addition, the noise distribution is symmetric. We can combine every two samples in a batch using this approach, which only reduces the batch size of each batch by a constant factor of 1/2. Thus, the assumption of symmetric noise can be eliminated by increasing the required batch sizes in our theorems by a factor of 2.

## K  Algorithm's Parameter Requirements

The algorithm requires two distribution parameters: the L4-L2 hypercontractivity parameter $C$ and the condition number of input covariance matrix $C_1$. These parameters significantly generalize the input distributions typically considered in mixed linear regression, which have primarily focused on sub-Gaussian distributions with an identity covariance matrix. Specifically, the L4-L2 hypercontractivity parameter generalizes the sub-gaussian distribution assumption and allows for heavier-tailed distributions. The condition-number $C_1$ of the input covariance matrices generalizes the identity covariance matrix assumption for input distribution, corresponding to $C_1 = 1$.

Additionally, the algorithm requires two population parameters: the number of sub-populations $k$ and the fraction of batches $\alpha_s$ present from the relevant sub-population. Note that the algorithm combines these parameters to define a single parameter $\ell$.

Finally, the algorithm requires two regression parameters: the length $M$ of the regression vector, and the additive noise variance $\sigma$.

In total the algorithm relies on six parameters, two of which can be combined into one. Although this may seem large, it requires only an upper bound on their values. If the upper bounds are tight, they will result in a good estimate. If for any parameter we do not have a tight upper bound, we can run the algorithm for different values of the parameter, dividing its possible range logarithmically. This approach increases the size of the resulting list at most logarithmically, ensuring that at least one estimate in the list will be tight.

## L  More Simulation Details

**Setup.** We have sets $B_s$ and $B_m$ of small and medium size batches and $k$ distributions $\mathcal{D}_i$ for $i \in \{0, 1, \ldots, k-1\}$. For a subset of indices $I \subseteq \{0, 1, \ldots, k-1\}$, both $B_s$ and $B_m$ have a fraction

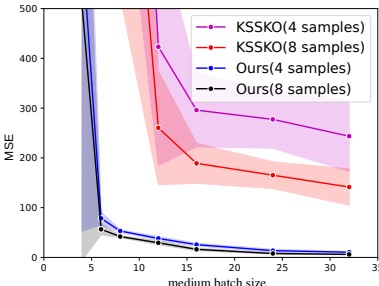
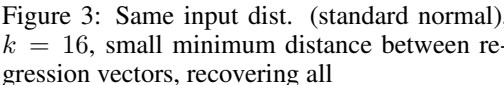
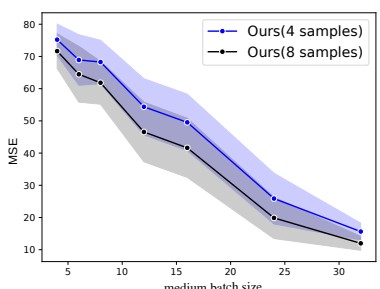

Figure 3: Same input dist. (standard normal), $k = 16$, small minimum distance between regression vectors, recovering all

Figure 4: Different input dist, $k = 100$, large minimum distance between regression vectors, recovering 4 components that have $1/16$ fraction of batches each

of $\alpha$ batches that contain i.i.d. samples from $\mathcal{D}_i$ for each $i \in I$. And for each $i \in \{0, 1, \ldots, k-1\} \setminus I$ in the remaining set of indices, $B_s$ and $B_m$ have $(1 - |I|/16)/(k - |I|)$ fraction of batches, that have i.i.d samples from $\mathcal{D}_i$. In all figures the output noise is distributed as $\mathcal{N}(0, 1)$.

All small batches have 2 samples each, while medium-size batches have $n_m$ samples each, which we vary from 4 to 32, as shown in the plots. We fix data dimension $d = 100$, $\alpha = 1/16$, number of small batches to $|B_s| = \min\{8dk^2, 8d/\alpha^2\}$ and the number of medium batches to $|B_m| = 256$. In all the plots, we average our 10 runs.

**Evaluation.** Our objective is to recover a small list containing good estimates for the regression vectors of $\mathcal{D}_i$ for each $i \in I$. We compare our proposed algorithm's performance with that of the algorithm in [KSS+20]. We generate lists of regression vector estimates $L_{\text{Ours}}$ and $L_{\text{KSSKO}}$ using our algorithm and [KSS+20], respectively. Then, we create 1600 new batches, each containing $n_{new}$ i.i.d samples randomly drawn from the distribution $\mathcal{D}_i$, where for each batch, index $i$ is chosen randomly from $I$.

Each list enables the clustering of the new sample batches. To cluster a batch using a list, we assign it to the regression vector in the list that achieves the lowest mean square error (MSE) for its samples.

To evaluate the average MSE for each algorithm, for each clustered batch, we generate additional samples from the distribution that the batch was generated from and calculate the error achieved by the regression vector in the list that the batch was assigned to. We then take the average of this error over all sets. We evaluate both algorithms' performance for new batch sizes $n_{new} = 4$ and $n_{new} = 8$, as shown in the plots.

**Minimum distance between regression vectors.** Our theoretical analysis suggests that our algorithm is robust to the case when the minimum distance between the regression vectors are much smaller than their norms. In order to test this, in Figure 3, we generate half of the regression vectors with elements independently and randomly distributed in $U[9, 11]$, and the other half with elements independently and randomly distributed in $U[-11, -9]$. Notably, the minimum gap between the vectors, in this case, is much smaller than their norm. It can be seen that the performance gap between our algorithm and the one in [KSS+20] increases significantly as we deviate from the assumptions required for the latter algorithm to work.

**Number of different distributions.** Our algorithm can notably handle very large $k$ (even infinite) while still being able to recover regression vectors for the subgroups that represent sufficient fraction of the data. In the last plot, we set $k = 100$ and $I = \{0, 1, 2, 3\}$ to highlight this ability. In this case, the first four distributions each generate a $1/16$ fraction of batches, and the remaining 96 distributions each generate a $1/128$ fraction of batches. We provide the algorithm with one additional medium-size batch from $\mathcal{D}_i$ for each $i \in I$ for identification of a list of size $I$. The results are plotted in Figure 4, where we can see that the performance gets better with medium batch size as expected. Note that the algorithm in [KSS+20] cannot be applied to this scenario.

