# OpenReview forum: "Linear Regression using Heterogeneous Data Batches"
_NeurIPS.cc/2024/Conference — NeurIPS 2024 spotlight_

### Official Review · Reviewer_D4Ej · 2024-07-05

**Soundness:** 3
**Presentation:** 3
**Contribution:** 4
**Rating:** 7
**Confidence:** 4

**Summary:**

This paper addresses the regression problem in scenarios where heterogeneous data is collected from multiple sources, necessitating learning tasks on small batches of samples. The approach involves dividing the data sources into k subgroups with unknown distributions. This study advances the work of Kong et al. (2020) by introducing a gradient-based algorithm, which does not require the restrictive assumptions present in previous works, such as the assumption of isotropic Gaussian distributions for all k subgroups. The properties of the proposed solution are examined through both theoretical and numerical analyses.

**Strengths:**

The general quality of the paper is commendable, and it addresses all pertinent issues effectively. It adequately covers related works and provides a comprehensive perspective on the topic.

The main topic is an important research area with applications in a wide range of real-world projects. Existing works in this area often rely on several assumptions regarding distribution and batch sizes, which can limit their efficiency and accuracy. This paper offers a more general solution that does not depend on restrictive assumptions, thereby enhancing its applicability and robustness.

**Weaknesses:**

The main concerns about the paper include:

It has not been properly discussed how this solution allows the recovery of models for sub-populations with a significant fraction of batches when there are a large number of subgroups. This aspect is mentioned several times but not explained clearly.
In particular, the paper lacks a detailed explanation of how this work can handle large subgroups more effectively than the reference work by Kong et al. (2020).

While it can improve upon previous related works, a detailed complexity analysis is needed. The proposed work is more complex than other works, such as Kong et al. (2020), in terms of both theory and computations. Eliminating the effect of restrictive assumptions can raise the complexity.

Additionally, the experiments are not sufficient for this work. Many aspects should be checked, and only a comparison with one baseline cannot support the claims made in the paper. Several other works in this area can be used for comparison, even if they have been proposed only for specific cases.

In Figures 1 and 2, the differences between the errors of the baselines when k is small are not significant. It would be beneficial for the authors to provide a theoretical explanation for this observation. Understanding why the error rates converge or show minimal variance under these conditions can provide deeper insights into the behavior of the algorithms and the impact of the number of subgroups on their performance.

**Questions:**

See Weaknesses.

**Limitations:**

The limitations have been partially addressed in the paper.

---

> ### Author Rebuttal · Authors · 2024-08-07
>
> We thank the reviewer for many positive comments about our paper. Below we address the remaining comments of the reviewer one by one.
>
> 1. Note that even when $k$ is arbitrarily large, the number of batches and batch sizes required in Theorem 2.1 and Corollary 2.2 are reasonable as in that case they depend on $\alpha$ instead of $k$, implying that batches can be recovered even for arbitrary large $k$. We also provide numerical experiments to further confirm this fact (see Figure 4 in the appendix.) Our main observation that helps in such a recovery for large $k$ is that even when $k=\infty$, a small subspace can be estimated that contains gradients for all components with a significant fraction present.
>
> 2. As discussed in line 112 of section 1.2, our sample complexity is lower than Kong et al. (2020). Our time complexity $\tilde O(|B_s| d^2/\alpha_m)$ is at most $\tilde O(1/\alpha_m)$ times the complexity of Kong et al. (2020). But as is clear from the value and our numerical experiments show, this complexity is still reasonable in practice.
>
>
> 3. Note that of all previous works, only Kong et al. (2020)  take advantage of the batch structure. All other works mentioned in the paper do not, and hence they will naturally fare worse than the baseline we considered. For that reason we did not include them in our results.
>
> 4. We note that in Figures 1 and 2, $k$ is kept constant at 16. The small gap between the two algorithms for small values of medium batch size is because if it is too small then it falls below the requirement of Theorem 2.1. In that case, both our algorithm and baselines fail to recover the regression vector.

---

> > ### Comment · Reviewer_D4Ej · 2024-08-12
> > **Official Comment**
> >
> > Thank you for addressing the feedback provided. After reviewing the revisions, I am pleased with the improvements and will be raising my score accordingly.

---

### Official Review · Reviewer_pdPW · 2024-07-08

**Soundness:** 4
**Presentation:** 2
**Contribution:** 3
**Rating:** 7
**Confidence:** 3

**Summary:**

The paper considers the problem of linear regression with heterogeneous data batches, where a user receives a collection of batches of data with not necessarily the same underlying signal distribution and they must attempt to infer a list of heterogeneous signals present in this data. This paper proposes a novel gradient-based method to approach this problem that attains provable and favourable guarantees with less assumptions than previous work.

**Strengths:**

Originality and significance: The paper considers a novel way to tackle the problem of linear regression with heterogeneous batches. The work is significant as their algorithm removes a number of assumptions in prior literature, and seems more performant than previous algorithms. The theory provided is solid and sound. The idea is good, and the description of the proof method and the grad clipping idea was good.

Quality and clarity: The problem is well motivated, and the contextualization with respect to prior works is made clear.

**Weaknesses:**

Clarity:

- There are multiple spelling mistakes: a) “liner” above section header 1.1, b) “which follows [a] linear regression model” in the middle of page 2, c) “liner” on the last paragraph of page 3, and many others. Please re-read the paper carefully for spelling mistakes.
- Theorem 2.3: The first sentence does not make sense to me. What does it mean?

More comments:

- Missing reference to recent novel developments in MLR: https://proceedings.mlr.press/v206/tan23a.html. The sample complexity in this work is not super-polynomial, but is indeed linear, and this should be mentioned, despite it still having to satisfy assumptions 2 and 4.
- I think one drawback of the work has not been mentioned when compared to MLR: you output a list of size L, which is potentially much bigger than the number of signals in the original data, and this should be mentioned. This relates to the problem of list-decoding, and it would be good to also mention that literature and how this relates: https://arxiv.org/abs/1905.05679.
- I appreciate that multiple assumptions have been removed, but also importantly one property of the solution has changed from MLR: you are content now with a list of possible signals. From this, it is less surprising that assumptions can be removed, as now the assumptions required and the problem formulation is closer overall to list-decodable linear regression. Could you comment on this?
- The input distributions can be different, but they must be known? Whether yes or no, I would mention this clearly in the contribution.
- I think the work could benefit from a better discussion on what happens to the algorithm when to regression vectors have no separation? How does the algorithm deal with that? Do we expect the size of the outputted list to change?
- A real data experiment would be nice, to evidence this work as that of potential practical significance, but this is not necessary and I still find the work strong and a significant contribution without it.

**Questions:**

What are the barriers to implementing/trying this on real data?

**Limitations:**

The authors have addressed most limitations in the work, I acknowledge the last paragraph of the discussions. Further addressing the comments above would help as well.

---

> ### Author Rebuttal · Authors · 2024-08-07
>
> We thank the reviewer for the careful review and positive comments. The reviewer’s comments are addressed herein.
>
> 1. We thank the reviewer for pointing out the typos, and will correct them along with some others we found.
>
> 2. The condition part of Theorem 2.3 can be more clearly stated as follows:
> Let index $i$ be in set $I$ and let list $L$ be a list of $d$ dimensional vectors. Suppose for a given $\beta>0$ at least one of the vectors in $L$ is within distance $\beta$ from the regression vector of distribution $\mathcal D_i$, namely $\min_{w\in L} ||w-w_i||\le \beta$.
>
> 3. We thank the reviewer for the reference to the recent work on mixed linear regression. It is clearly relevant, and we will include it in our paper.
>
>  4. (see next point)
> 5. Addressing 4 and 5 together, note that, as stated in corollary 2, the $L$ identified by our algorithm is at most $\tilde O(1/\alpha_m)$, even for arbitrary large $k$.
> Furthermore, if all $k$ sub-populations follow the assumption in Theorem 2.1 the argument preceding corollary 2.2 can be modified as follows to show that the list $L$ identified by our algorithm can be trimmed down to have at most $k$ elements. In such a case, for any choice of $b^*$ from $B_m$ to run algorithm 1, the regression vector estimates $\hat w$ will satisfy $||\hat w-w_i||\le \epsilon \sigma$ for some $i\in [k]$.
> A simple argument using the triangle inequality shows that removing the estimates from $L$ until no two estimates are within distance $2\epsilon\sigma$ from the list $L$ will result in a list $L$ of size $\le k$ such that for all $i$ we have  $||w_i-\hat w||\le 3\epsilon \sigma$.
> Also, we note that we already included the literature on list decoding. We referred to a recent work of list decodable linear regression using batches [DJKS22] in section 1.2 and other references in the appendix. We can discuss these connections in more detail as per the reviewer’s suggestion.
>
> 6. The input distributions need not be known. We will mention it clearly in the contributions following your suggestion.
>
> 7. If for two distributions the regression vectors have no separation then when the algorithm is run using a medium-sized batch as $b^*$ from either of two distributions we will be estimating the same regression vector. The upper bound on the size of the output list is independent of the separation.
>
> 8. As shown by the synthetic experiments in Figures 1-4, the algorithm is practical even for reasonable-size datasets and dimension $d$. As such there are no barriers to trying it out on real data. We leave it to future work to find interesting and relevant datasets for this setting.

---

> > ### Comment · Reviewer_pdPW · 2024-08-10
> > **Response**
> >
> > Dear authors,
> >
> > Thank you for addressing my questions, I am happy with the answers and have increased my rating accordingly.

---

### Official Review · Reviewer_oZju · 2024-07-11

**Soundness:** 3
**Presentation:** 4
**Contribution:** 3
**Rating:** 7
**Confidence:** 4

**Summary:**

This paper explores the concurrent learning of multiple linear models from various small batches of input-output pairs, each derived from a distinct, small-sized source dataset. Although there can be a large number of sources, it is assumed that each dataset belongs to one of several fixed but unknown subgroups, where each subgroup follows a specific input-output rule. These rules may not be linear for all subgroups; however, a significant number of subgroups, representing a substantial portion of the datasets, follow separate linear rules.

The paper provides theoretical guarantees that the linear models for most datasets can be recovered, even with minimal data in each dataset. Moreover, the algorithm's computational complexity is polynomial. The paper claims considerable improvements over existing works, with a detailed comparison with prior research in Section 1.2. The authors have detailed their algorithm, which (similar to some prior works) is based on "subspace estimation," in Section 3. Additionally, the paper includes a series of computational experiments, although I have not thoroughly examined them. I have not fully reviewed the proofs regarding the nature and efficiency of the proposed algorithm.

I would like to see other reviewers' comments. However, my current assessment is that this paper is a good candidate for NeurIPS. I vote for acceptance.

**Strengths:**

The paper is well-written and well-motivated. The problem addressed is natural and aligns with numerous modern applications in machine learning and data science. The review of prior works is thorough, and the authors have effectively positioned their work within the existing literature.

The authors claim significant advancements in their bounds concerning the relationship to certain parameters and dimensions compared to previous works. For example, their bounds require at least $\Omega\left(k^{3/2}\right)$ batches of data, where each batch contains at least $\Omega\left(k^{1/2}/\epsilon^2\right)$ samples. As can be seen, the number of batches is independent of $\epsilon$ (the estimation error). In comparison, some of the most notable prior works require the number of batches to grow significantly with $\epsilon$.

The authors have mitigated several significant constraints present in prior works, including eliminating the assumption of a Gaussian distribution for feature vectors. Additionally, this work removes the assumption that all subgroups must follow a linear model. In fact, the authors extend the concept to accommodate scenarios where the number of subgroups can even approach infinity, ensuring the recovery of the "majority" of subgroups with linear models.

**Weaknesses:**

There are too many hyper-parameters in Algorithm 1, which are provided to the algorithm as input. However, I found the discussion on how to tune these hyper-parameters in practice to be lacking. It would be beneficial to include more detailed guidelines or heuristics for selecting appropriate values for these hyper-parameters, potentially based on empirical studies or theoretical insights. Additionally, some of the assumptions in Section 2.1, such as the distributional properties of the input data and the specific conditions under which the theoretical guarantees hold, may need more clarification. Providing concrete examples or scenarios illustrating these assumptions would help in understanding their practical implications and ensuring they are not overly restrictive.

**Questions:**

What are the main intuitions behind the assumptions made in Remark 2.1? (L. 196)

**Limitations:**

-

---

> ### Author Rebuttal · Authors · 2024-08-07
>
> We thank the reviewer for many positive comments. The reviewer’s main concerns involve the number of parameters and distributional assumptions, both addressed below.
>
> While the number of parameters may seem large, each has a natural and important contribution.
>
> Two accuracy parameters $\epsilon$ and $\delta$ that are part and parcel of any PAC learning algorithm. They specify the accuracy we are looking for and determine the number of samples needed to achieve it.
>
> Two regression parameters: the length $M$ of the regression vector, and the additive noise variance $\sigma$, reflect the complexity of the input-output relation.
>
> Two population parameters: the number $k$ of sub-populations and the fraction of batches $\alpha$ present from the relevant sub-population.
> Note that the algorithm combines these parameters to set one parameter $\ell$.
>
> Finally, two distribution parameters: the L4-L2 hypercontractivity parameter $C$ and the condition number of input covariance matrix $C_1$. These parameters reflect the complexity/richness of the distribution class and substantially generalize input distributions considered in the past for mixed linear regression, as they only considered sub-gaussian distributions with the identity covariance matrix.
>
>
> The L4-L2 hypercontractivity parameter significantly generalizes the sub-gaussian distribution assumption and allows for heavier-tailed distributions.
> The condition-number $C_1$ of the input covariance matrices significantly generalizes the identity covariance matrix assumption for input distribution that corresponds to $C_1=1$.
>
> Though the number of parameters seems large, the algorithm needs only an upper bound on their values. If the upper bounds are tight, they will result in a good estimate.
> If for any parameter we do not have a tight upper bound, we can run the algorithm for different values of the parameters, dividing their possible range logarithmically, and
> choose the parameters that achieve a small error on a holdout set.
> We can add this approach in more detail in the final version of the paper and formally define error computation and selection for our setting.
>
> Furthermore, it is easy to see that these upper bounds are needed. For example, even for the simplest case when all samples are from the same sub-population, some condition on the anti-concentration of  $XX^T$ as above is needed to obtain finite sample complexities [Lecué and Mendelson, 2016, Oliveira, 2016] in heavy-tailed linear regression, and this corresponds to a low condition number.
>
> The reviewer also asks about the intuition behind Remark 2.1. Our algorithm and its analysis are based on gradient descent, and can handle stochastic noise. Therefore we require only the distribution of gradients to be similar for batches in the same sub-population and since the analysis can tolerate statistical noise it can also tolerate deviation in the distribution of gradients across batches as long as it is small.

---

> > ### Comment · Reviewer_oZju · 2024-08-11
> > **Response to rebuttal**
> >
> > I would like to thank the authors for addressing my questions. I have no further concerns and will maintain my score and support for this work.

---

### Official Review · Reviewer_Xj2G · 2024-07-13

**Soundness:** 4
**Presentation:** 3
**Contribution:** 3
**Rating:** 7
**Confidence:** 3

**Summary:**

The paper presents an algorithm that handles a situation of linear regression from many heterogeneous batches of data, and simultaneously learns all linear separators for which there are large enough batches of data witnessing them. The actual guarantees are more nuanced, covering the common case of a heavy-tailed batch size distribution with completely unknown and potentially difficult-to-distinguish batches.

**Strengths:**

There's a lot to like about the overall approach to the problem. I would compliment it as being "applied theory," in contrast to a lot of prior work which takes a more purely theoretical assumption-laden mindset (as commented upon also by the authors, in describing the many assumptions that they are able to remove). Algorithmic techniques like clipping and median-finding are coupled with appropriate theoretical choices, to achieve a sample complexity that's nearly optimal for these regimes.

The applicability of the problem itself is more than meets the eye; theory can motivate usage of certain algorithmic techniques. In this case, working in the dual gradient domain to do clipping has a very helpful effect, as the authors prove -- and this is currently not standard for the problem.

**Weaknesses:**

Sec. 1.2 and 1.1 cover overlapping material, and in addition they could be switched with each other. I found 1.2 much clearer about the top-level contributions (e.g. removal of assumptions), and 1.1 more of a detailed theoretical comparison. Overall, some rearrangements of the initial section 1 would improve presentation (e.g. "Our work" line 34 unclear reference at that point for the reader). Would suggest moving forward "improvement over prior work" subsection even within Sec. 1.2. It takes 3.5 pages to formally define the setting with batches, continuing the reader's ambiguity somewhat.

For an paper that uses some nice applied algorithmic techniques (see Strengths), it would be nice to run it in a less contrived manner; the R-way partitioning of the data seriously affects applicability (see Questions).

**Questions:**

There remains a significant gap between the scenario analyzed here and a practical one. The number of GD steps R is a real bottleneck because it requires partitioning the data R ways. Though I understand the authors saying "this division may not be necessary for practical implementation," there are certainly extreme datasets in which it would and would not be necessary. It seems to me like this may be just an analysis consideration, and a more refined analysis might highlight under what stability-type conditions the partitioning would be needed. This is significant; a large R makes it possible to get some of the nice adaptivity properties e.g. to \Delta .

Duchi et al. have work on Ergodic GD which might be of interest for its techniques in this matter, and its parametrizations; doubtless there is more recent follow-up work in this line. Would appreciate comment from the authors on this issue of R.

**Limitations:**

Yes.

---

> ### Author Rebuttal · Authors · 2024-08-07
>
> We thank the reviewer for the positive review, the appreciation of our contribution, and the two suggestions that we discuss next.
>
> The first suggestion concerns rearranging Sections 1.1 and 1.2. Section 1.1 discusses the paper’s main results, and section 1.2 addresses the results’ improvement over prior work. We agree with the reviewer that the comparison in Section 1.2 results in some duplication of results outlined in Section 1.1, and will rearrange some of the material to minimize the repetition.
>
> The second comment concerns R-way partitioning of the data. More concretely, our algorithm runs in R rounds, and our theoretical analysis requires the use of independent samples in each round.
>
> We thank the reviewer for mentioning that Duchi’s work might help eliminate the need for this R-way partitioning in our analysis either entirely or under appropriate constraints.
>
> In searching papers on Ergodic Gradient Descent by Duchi et. al. we found the paper “Ergodic Mirror Descent” that we believe the reviewer referred to. While this paper performs gradient descent using ergodic samples, and hence generalizes the independent samples we use, it still does not eliminate the need of new samples in each round of gradient descent. In the limited rebuttal period, we could not find a way to extend the analysis in the paper to allow using the same samples in each algorithm round, and we leave that for future work.
>
> Note however that to achieve estimation accuracy $\epsilon \sigma$ with noise variance $\le \sigma^2$, and regression vectors of length at most $M$, the number of rounds $R$ required by the algorithm is just $\log(M/\sigma)$. Hence the factor $R$ increase in the sample complexity implied by our theoretical analysis is at most a logarithmic factor.
> Furthermore, in our numerical experiments, the algorithm recovers the regression vectors even when the same samples are used in each round. However, we agree with the reviewer that without a theoretical proof it is unclear if it will not be necessary for some extreme datasets.

---

> > ### Comment · Reviewer_Xj2G · 2024-08-12
> >
> > Thanks to the authors for addressing my questions, and points about the presentation. I will maintain my score and positive review.

---

### Official Review · Reviewer_ntyP · 2024-07-14

**Soundness:** 4
**Presentation:** 3
**Contribution:** 3
**Rating:** 7
**Confidence:** 3

**Summary:**

The paper proposes an algorithm to solve linear regression problems from batched data. The regression coefficients for each batch are assumed to be heterogeneous but come from $k$ possible values. The algorithm proposes a few novel approaches to identify which medium-sized batches are close to a given batch. It also improves the statistical error by estimating a low-rank subspace from small batches. The knowledge-integration techniques developed in the paper are intriguing, and the numerical performance is better than [KSS+20].

**Strengths:**

The introduction of the algorithm is mostly clear, despite its complexity. The intuitions are well-explained. The gradient clipping technique is novel compared with [KSS+20]. Also, the use of gradient descent in the algorithm presents an interesting contribution.

**Weaknesses:**

In line 123, the authors claim that the model does not rely on the previous assumptions. Authors can also briefly discuss how their algorithm innovations circumvents these assumptions.



Some minor issues about writing:

The sentence in line 332 is very long and hard to follow.

In line 836, there is an error.

**Questions:**

The number of small batches required scales with $d$, which is relatively large in high dimensions. Can authors give specific real-life examples to justify the assumptions on $|B_s|$ and $|B_m|$ in Theorem 2.1?

Authors claim that $k$ can be infinity, as long as the constraints of $\alpha$ are satisfied. Do authors test the claim numerically?

**Limitations:**

See weakness.

---

> ### Author Rebuttal · Authors · 2024-08-07
>
> Thank you for the careful review and constructive feedback, and for appreciating the contribution, novelty, presentation clarity, and intuition.
>
> Regarding your comments and questions:
>
> Line 123: Section 1.3 summarizes the algorithm’s innovations, and we will elaborate on how they help overcome assumptions in prior works.
>
> Line 332: We will break it into several sentences so it is easier to follow.
>
> Line 836: We will add the missing reference.
>
> Next, we answer the two questions raised in the review.
>
> For an example motivating the number of small batches, observe that in many recommendation systems, most users rate only a few items. The ratings by any such user may serve as a small batch. Since these systems have many users, the number of small batches will also be large. Observe also that typically relatively few users provide a fair number of ratings, and they can serve as medium-sized batches.
>
> For a test justifying that $k$ can be large, Figure 4 in the appendix shows an experiment where the number $k$ of subpopulations is large (100), while the number of subpopulations with sufficiently many batches for enabling recovery is small (4). Our theoretical analysis shows that this experiment can be repeated with even larger $k$, without impacting the algorithm’s performance.

---

### Decision · Program_Chairs · 2024-09-25

**Decision:**

Accept (spotlight)

**Comment:**

Five reviewers evaluated the paper, and their overall assessment is positive. I agree with their evaluation and believe the paper makes a strong contribution with compelling results. The reviewers particularly appreciated that the paper addresses an important topic—linear regression with heterogeneous data—and delivers theoretically sound results without relying on unrealistic assumptions on the distribution of the data. Both the reviewers and the AC believe that this is one of the first papers that can handle inputs with non-isotropic and heavy-tailed in heterogeneous settings. Moreover, the paper is exceptionally well-written and easy to read. Overall, the AC holds a very positive opinion of this work.